# Mitigating Statistical Bias within Differentially Private Synthetic Data

**Sahra Ghalebikesabi**[1] **Harrison Wilde**[2] **Jack Jewson**[3] **Arnaud Doucet**[1]

**Sebastian Vollmer**[5] **Chris Holmes**[1]

[1]University of Oxford
[2]University of Warwick
[3]Universitat Pompeu Fabra
[5]University of Kaiserslautern, German Research Centre for Artificial Intelligence (DFKI)

## Abstract

Increasing interest in privacy-preserving machine learning has led to new and evolved approaches for generating private synthetic data from undisclosed real data. However, mechanisms of privacy preservation can significantly reduce the utility of synthetic data, which in turn impacts downstream tasks such as learning predictive models or inference. We propose several re-weighting strategies using privatised likelihood ratios that not only mitigate statistical bias of downstream estimators but also have general applicability to differentially private generative models. Through large-scale empirical evaluation, we show that private importance weighting provides simple and effective privacy-compliant augmentation for general applications of synthetic data.

## 1 INTRODUCTION

The prevalence of sensitive datasets, such as electronic health records, contributes to a growing concern for violations of an individual's privacy. In recent years, the notion of Differential Privacy (Dwork et al., 2006) has gained popularity as a privacy metric offering statistical guarantees. This framework bounds how much the likelihood of a randomised algorithm can differ under neighbouring real datasets. We say two datasets $\mathcal{D}$ and $\mathcal{D}'$ are neighbouring when they differ by at most one observation. A randomised algorithm $g : \mathcal{M} \to \mathcal{R}$ satisfies $(\epsilon, \delta)$-differential privacy for $\epsilon, \delta \geq 0$ if and only if for all neighbouring datasets $\mathcal{D}, \mathcal{D}'$ and all subsets $S \subseteq \mathcal{R}$, we have

$$\Pr(g(\mathcal{D}) \in S) \leq \delta + e^\epsilon \Pr(g(\mathcal{D}') \in S).$$

The parameter $\epsilon$ is referred to as the privacy budget; smaller $\epsilon$ quantities imply more private algorithms.

Injecting noise into sensitive data according to this paradigm allows for datasets to be published in a private manner. With the rise of generative modelling approaches, such as Generative Adversarial Networks (GANs) (Goodfellow et al., 2014), there has been a surge of literature proposing generative models for differentially private (DP) synthetic data generation and release (Jordon et al., 2019; Xie et al., 2018; Zhang et al., 2017). These generative models often fail to capture the true underlying distribution of the real data, possibly due to flawed parametric assumptions and the injection of noise into their training and release mechanisms. The constraints imposed by privacy-preservation can lead to significant differences between nature's true data generating process (DGP) and the induced synthetic DGP (SDGP) (Wilde et al., 2020). This increases the bias of estimators trained on data from the SDGP which reduces their utility.

Recent literature has proposed techniques to decrease this bias by modifying the training processes of private algorithms. These approaches are specific to a particular synthetic data generating method (Zhang et al., 2018; Frigerio et al., 2019; Neunhoeffer et al., 2020), or are query-based (Hardt and Rothblum, 2010; Liu et al., 2021) and are thus not generally applicable. Hence, we propose several post-processing approaches that aid mitigating the bias induced by the DP synthetic data.

While there has been extensive research into estimating models directly on protected data without leaking privacy, we argue that releasing DP synthetic data is crucial for rigorous statistical analysis. This makes providing a framework to debias inference on this an important direction of future research that goes beyond the applicability of any particular DP estimator. Because of the post-processing theorem (Dwork et al., 2014), any function on the DP synthetic data is itself DP. This allows deployment of standard statistical analysis tooling that may otherwise be unavailable for DP estimation. These include 1) exploratory data analysis, 2) model verification and analysis of model diagnostics, 3) private release of (newly developed) models for which no DP analogue has been derived, 4) the computation of con-

*Accepted for the 38th Conference on Uncertainty in Artificial Intelligence (UAI 2022).*

fidence intervals of downstream estimators through the non-parametric bootstrap, and 5) the public release of a data set to a research community whose individual requests would otherwise overload the data curator. This endeavour could facilitate the release of data on public platforms like the UCI Machine Learning Repository (Lichman, 2013) or the creation of data competitions, fuelling research growth for specific modelling areas.

This motivates our main contributions, namely the formulation of multiple approaches to generating DP importance weights that correct for synthetic data's issues. In particular, this includes:

- The bias estimation of an existing DP importance weight estimation method, and the introduction of an unbiased extension with smaller variance (Section 3.3).

- An adjustment to DP Stochastic Gradient Descent's sampling probability and noise injection to facilitate its use in the training of DP-compliant neural network-based classifiers to estimate importance weights from combinations of real and synthetic data (Section 3.4).

- The use of discriminator outputs of DP GANs as importance weights that do not require any additional privacy budget (Section 3.5).

- An application of importance weighting to correct for the biases incurred in Bayesian posterior belief updating with synthetic data motivated by the results from (Wilde et al., 2020) and to exhibit our methods' wide applicability in frequentist and Bayesian contexts (Section 3.1).

## 2 BACKGROUND

Before we proceed, we provide some brief background on bias mitigation in non-private synthetic data generation.

### 2.1 DENSITY RATIOS FOR NON-PRIVATE GANS

Since their introduction, GANs have become a popular tool for synthetic data generation in semi-supervised and unsupervised settings. GANs produce realistic synthetic data by trading off the learning of a generator $Ge$ to produce synthetic observations, with that of a classifier $Di$ learning to correctly classify the training and generated data as real or fake. The generator $Ge$ takes samples from the prior $u \sim p_u$ as an input and generates samples $Ge(u) \in X$. The discriminator $Di$ takes an observation $x \in X$ as input and outputs the probability $Di(x)$ of this observation being drawn from the true DGP. The classification network $Di$ distinguishes between samples from the DGP with label $y = 1$ and distribution $p_D$, and data from the SDGP with label $y = 0$ and distribution $p_G$. Following Bayes' rule we can show that the output of $Di(x)$, namely the probabilities $\widehat{p}(y = 1|x)$ and

$\widehat{p}(y = 0|x)$, can be used for importance weight estimation:

$$\frac{\widehat{p}_D(x)}{\widehat{p}_G(x)} = \frac{\widehat{p}(x|y = 1)}{\widehat{p}(x|y = 0)} = \frac{\widehat{p}(y = 1|x)}{\widehat{p}(y = 0|x)} \frac{\widehat{p}(y = 0)}{\widehat{p}(y = 1)}. \quad (1)$$

This observation has been exploited in a stream of literature focusing on importance weighting (IW) based sampling approaches for GANs. Grover et al. (2019) analyse how importance weights of the GAN's outputs can lead to performance gains; extensions include their proposed usage in rejection sampling on the GAN's outputs (Azadi et al., 2018), and Metropolis–Hastings sampling from the GAN alongside improvements to the robustness of this sampling via calibration of the discriminator (Turner et al., 2019). To date, no one has leveraged these discriminator-based IW approaches in DP settings where the weights can mitigate the increased bias induced by privatised data models.

### 2.2 DIFFERENTIAL PRIVACY IN SYNTHETIC DATA GENERATION

Private synthetic data generation through DP GANs is built upon the post processing theorem: If $Di$ is $(\epsilon, \delta)$- DP, then any composition $Di \circ Ge$ is also $(\epsilon, \delta)$-DP (Dwork et al., 2014) since $Ge$ does not query the protected data. Hence, to train private GANs, we only need to privatise the training of their discriminators, see e.g. Hyland et al. (2018). Xie et al. (2018) propose DPGAN, a Wasserstein GAN which is trained by injecting noise to the gradients of the discriminator's parameters. In contrast, Jordon et al. (2019) privatise the GAN discriminator by using the Private Aggregation of Teacher Ensembles algorithm. Recently, Torkzadehmahani et al. (2019) proposed DPCGAN as a conditional variant to DPGAN that uses an efficient moments accountant. In contrast, PrivBayes (Zhang et al., 2017) learns a DP Bayesian network and does not rely on a GAN-architecture. Other generative approaches, for instance, include Chen et al. (2018); Acs et al. (2018). See Abay et al. (2018); Fan (2020) for an extensive overview of more DP generative approaches.

**Differentially private bias mitigation** In this paper, we offer an augmentation to the usual release procedure for synthetic data by leveraging true and estimated importance weights. Most related to our work are the contributions from Elkan (2010) and Ji and Elkan (2013) who train a regularised logistic regression model and assign weights based on the Laplace-noise-contaminated coefficients of the logistic regression. In follow up work, Ji et al. (2014) propose to modify the update step of the Newton-Raphson optimisation algorithm used in fitting the logistic regression classifier to achieve DP. However, neither of these generalise well to more complex and high dimensional settings because of the linearity of the classifier. Further, the authors assume the existence of a *public dataset* while we consider the case where we first generate DP *synthetic data* and then weight them a posteriori, providing a generic and universally

applicable approach. The benefit of learning a generative model over using public data include on the one hand that there is no requirement for the existence of a public data set, and on the other hand the possibility to generate new data points. This distinction necessitates additional analysis as the privacy budget splits between the budget spent on fitting the SDGP and the budget for estimating the IW approach. Furthermore, we show that the approach from Ji and Elkan (2013) leads to statistically biased estimation and formulate an unbiased extension with improved properties.

## 3 DIFFERENTIAL PRIVACY AND IMPORTANCE WEIGHTING

From a decision theoretic perspective, the goal of statistics is estimating expectations of functions $h : X \mapsto \mathbb{R}$, e.g. loss or utility functions, w.r.t the distribution of future uncertainties $x \sim p_D$. Given data from $\{x'_1, \ldots, x'_{N_D}\} =: x'_{1:N_D} \overset{i.i.d.}{\sim} p_D$ the data analyst can estimate these expectations consistently via the strong law of large numbers as $\mathbb{E}_{x \sim p_D}(h(x)) \approx \frac{1}{N_D} \sum_{i=1}^{N_D} h(x'_i)$. However, under DP constraints the data analyst is no longer presented with a sample from the true DGP $x'_{1:N_D} \overset{i.i.d.}{\sim} p_D$ but with a synthetic data sample $x_{1:N_G}$ from the SDGP $p_G$. Applying the naive estimator in this scenario biases the downstream tasks as $\frac{1}{N_G} \sum_{i=1}^{N_G} h(x_i) \to \mathbb{E}_{x \sim p_G}(h(x))$ almost surely.

This bias can be mitigated using a standard Monte Carlo method known as importance weighting (IW). Suppose we had access to the weights $w(x) := \frac{p_D(x)}{p_G(x)}$. If $p_G(\cdot) > 0$ whenever $h(\cdot)p_D(\cdot) > 0$, then IW relies on

$$\mathbb{E}_{x \sim p_D}[h(x)] = \mathbb{E}_{x \sim p_G}[w(x)h(x)]. \quad (2)$$

So we have almost surely for $x_{1:N_G} \overset{i.i.d.}{\sim} p_G$ the convergence

$$I_N(h|w) := \frac{1}{N_G} \sum_{i=1}^{N_G} w(x_i)h(x_i) \overset{N_G \to \infty}{\longrightarrow} \mathbb{E}_{x \sim p_D}[h(x)].$$

### 3.1 IMPORTANCE WEIGHTED EMPIRICAL RISK MINIMISATION

A downstream task of particular interest is the use of $x'_{1:N_D} \sim p_D$ to learn a predictive model, $f(\cdot) \in \mathcal{F}$, for the data generating distribution $p_D$ based on empirical risk minimisation. Given a loss function $h : \mathcal{F} \times X \mapsto \mathbb{R}$ comparing models $f(\cdot) \in \mathcal{F}$ with observations $x \in X$ and data $x'_{1:N_D} \sim p_D$, the principle of empirical risk minimisation (Vapnik, 1991) states that the optimal $\widehat{f}$ is given by the minimisation of

$$\frac{1}{N_D} \sum_{i=1}^{N_D} h(f(\cdot), x'_i) \approx \mathbb{E}_{x \sim p_D}[h(f(\cdot), x)]$$

over $f$. Maximum likelihood estimation (MLE) is a special case of the above with $h(f(\cdot), x_i) = -\log f(x_i|\theta)$ for a class of densities $f$ parameterised by $\theta$. Given synthetic data $x_{1:N_G} \sim p_G$, Equation (2) can be used to debias the learning of $f$.

**Remark 1** (Supplement B.5). *Minimisation of the importance weight adjusted log-likelihood, $-w(x_i) \log f(x_i|\theta)$, can be viewed as an $M$-estimator (e.g. Van der Vaart, 2000) with clear relations to the standard MLE.*

**Bayesian Updating.** Wilde et al. (2020) showed that naively conducting Bayesian updating using DP synthetic data without any adjustment could have negative consequences for inference. To show the versatility of our approach and to address the issues they pointed out, we demonstrate how IW can help mitigate this. The posterior distribution for parameter $\theta$ given $\tilde{x}' := x'_{1:N_D} \sim p_D$ is

$$\pi(\theta|\tilde{x}') \propto \pi(\theta) \prod_{i=1}^{N_D} f(x'_i|\theta) = \pi(\theta) \exp\left(\sum_{i=1}^{N_D} \log f(x'_i|\theta)\right)$$

where $\pi(\theta)$ denotes the prior distribution for $\theta$. This posterior is known to learn about model parameter $\theta_{p_D}^{KLD} := \arg\min_\theta KLD(p_D||f(\cdot|\theta))$ (Berk, 1966; Bissiri et al., 2016) where KLD denotes the Kullback-Leibler divergence.

Given only synthetic data $\tilde{x} := x_{1:N_G}$ from the 'proposal distribution' $p_G$, we can use the importance weights defined in Equation (2) to construct the (generalised) posterior distribution

$$\pi_{IW}(\theta|\tilde{x}) \propto \pi(\theta) \exp\left(\sum_{i=1}^{N_G} w(x_i) \log f(x_i|\theta)\right). \quad (3)$$

In fact, Equation (3) corresponds to a generalised Bayesian posterior (Bissiri et al., 2016) with $\ell_{IW}(x_i; \theta) := -w(x_i) \log f(x_i|\theta)$, providing a coherent updating of beliefs about parameter $\theta_{p_D}^{KLD}$ using only data from the SDGP.

**Theorem 1** (Supplement B.6). *The importance weighted Bayesian posterior $\pi_{IW}(\theta|x_{1:N_G})$, defined in Equation (3) for $x_{1:N_G} \overset{i.i.d.}{\sim} p_G$, admits the same limiting Gaussian distribution as the Bayesian posterior $\pi(\theta|x'_{1:N_D})$ where $x'_{1:N_D} \overset{i.i.d.}{\sim} p_D$, under regularity conditions as in (Chernozhukov and Hong, 2003; Lyddon et al., 2018).*

It is necessary here to acknowledge the existence of methods to directly conduct privatised Bayesian updating (e.g. Dimitrakakis et al., 2014; Foulds et al., 2016; Wang et al., 2015) or M-estimation (Avella-Medina, 2021). We refer the reader Section 1 for why the attention of this paper focuses on downstream tasks for private synthetic data. We consider the application of DP IW to Bayesian updating as a natural example of such a task.

## 3.2 ESTIMATING THE IMPORTANCE WEIGHTS

The previous section shows that IW can be used to recalibrate inference for synthetic data. Unfortunately, both the DGP $p_D$ and SDGP $p_G$ densities are typically unknown, e.g. due to the intractability of GAN generation, and thus the 'perfect' weight $w(x)$ cannot be calculated. Instead, we must rely on estimates of these weights, $\widehat{w}(x)$. In this section, we show that the existing approach to DP importance weight estimation is biased, and how the data curator can correct it.

Using the same reasoning as in Section 2.1, we argue that any calibrated classification method that learns to distinguish between data from the DGP, labelled thenceforth with $y = 1$, and from the SDGP, labelled with $y = 0$, can be used to estimate the likelihood ratio (Sugiyama et al., 2012). Using Equation (1), we compute

$$\widehat{w}(x) = \frac{\widehat{p}(y = 1|x)}{\widehat{p}(y = 0|x)} \frac{N_D}{N_G}$$

where $\widehat{p}$ are the probabilities estimated by such a classification algorithm. To improve numerical stability, we can also express the log weights as

$$\log \widehat{w}(x) = \sigma^{-1}(\widehat{p}(y = 1|x)) + \log \frac{N_D}{N_G},$$

where $\sigma(x) := (1 + \exp(-x))^{-1}$ is the logistic function and $\sigma^{-1}(\widehat{p}(y = 1|x))$ are the logits of the classification method. We will now discuss two such classifiers: logistic regression and neural networks.

## 3.3 PRIVATISING LOGISTIC REGRESSION

DP guarantees for a classification algorithm $g$ can be achieved by adding noise to the training procedure. The scale of this noise is determined by how much the algorithm differs when one observation of the dataset changes. In more formal terms, the sensitivity of $g$ w.r.t a norm $|\cdot|$ is defined by the smallest number $S(g)$ such that for any two neighbouring datasets $\mathcal{D}$ and $\mathcal{D}'$ it holds that

$$|g(\mathcal{D}) - g(\mathcal{D}')| \leq S(g).$$

Dwork et al. (2006) show that to ensure the differential privacy of $g$, it suffices to add Laplacian noise with standard deviation $S(g)/\epsilon$ to $g$.

Possibly the simplest classifier $g$ one could use to estimate the importance weights is logistic regression with $L_2$ regularisation. It turns out this also has a convenient form for its sensitivity. If the data is scaled to a range from 0 to 1 such that $X \subset [0, 1]^d$, Chaudhuri et al. (2011) show that the $L_2$ sensitivity of the optimal coefficient vector estimated by $\widehat{\beta}$ in a regularised logistic regression with model

$$\widehat{p}(y = 1|x_i) = \sigma(\widehat{\beta}^T x_i) = \left(1 + e^{-\widehat{\beta}^T x_i}\right)^{-1}$$

is $S(\widehat{\beta}) = 2\sqrt{d}/(N_D \lambda)$ where $\lambda$ is the coefficient of the $L_2$ regularisation term added to the loss during training. For completeness, when the logistic regression contains an intercept parameter, we let $x_i$ denote the concatenation of the feature vector and the constant 1.

Ji and Elkan (2013) propose to compute DP importance weights by training such an $L_2$ regularised logistic classifier on the private and the synthetic data, and perturb the coefficient vector $\widehat{\beta}$ with Laplacian noise. For a $d$ dimensional noise vector $\zeta$ with $\zeta_j \overset{i.i.d.}{\sim}$ Laplace$(0, \rho)$ with $\rho = 2\sqrt{d}/(N_D \lambda \epsilon)$ for $j \in \{1, \ldots, d\}$, the private regression coefficient is then $\overline{\beta} = \widehat{\beta} + \zeta$, akin to adding heteroscedastic noise to the private estimates of the log weights

$$\log \overline{w}(x_i) = \overline{\beta}^T x_i = \widehat{\beta}^T x_i + \zeta x_i. \tag{4}$$

The resulting privatised importance weights can be shown to lead to statistically biased estimation.

**Proposition 1** (Supplement B.1). *Let $\overline{w}$ denote the importance weights computed by noise perturbing regression coefficients as in Equation (4) (Ji and Elkan, 2013, Algorithm 1). The IS estimator $I_N(h|\overline{w})$ is biased.*

Introducing bias on downstream estimators of sensitive information is undesirable as it can lead to an increased expected loss. To address this issue, we propose a way for the data curator to debias the weights after computation.

**Proposition 2** (Supplement B.2). *Let $\overline{w}$ denote the importance weights computed by noise perturbing the regression coefficients as in Equation (4) (Ji and Elkan, 2013, Algorithm 1) where $\zeta$ can be sampled from any noise distribution that ensures $(\epsilon, \delta)$-differential privacy of $\overline{\beta}$. Define*

$$b(x_i) := 1/\mathbb{E}_{p_\zeta}[\exp\left(\zeta^T x_i\right)],$$

*and adjusted importance weight*

$$\overline{w}^*(x_i) = \overline{w}(x_i) b(x_i) = \widehat{w}(x_i) \exp\left(\zeta^T x_i\right) b(x_i). \tag{5}$$

*The importance sampling estimator $I_N(h|\overline{w}^*)$ is unbiased and $(\epsilon, \delta)$-DP for $\mathbb{E}_{p_\zeta}[\exp\left(\zeta^T x_i\right)] > 0$.*

In Supplement B.2.4, we further show that our approach does not only decrease the bias, but also the variance of the importance weighted estimators.

For the case of component-wise independent Laplace perturbations $\zeta_j \overset{i.i.d.}{\sim}$ Laplace$(0, \rho)$, we show that the bias correction term can be computed as

$$b(x_i) = \prod_{j=1}^{d} \left(1 - \rho^2 x_{ij}^2\right), \text{ provided } |x_{ij}| < 1/\rho \quad \forall j.$$

In practice, e.g. as we observe empirically in Section 4, the optimal choice of the regularisation term $\lambda$ is sufficiently

large such that $\rho < 1$. Since the data is scaled to a range of 0 to 1 (Chaudhuri et al., 2011), this bias correction method is not limited by the restriction $|x_{ij}| < 1/\rho, \forall j$. If the data curator still encounters a case where this condition is not fulfilled, they can choose to perturb the weights with Gaussian noise instead, in which case the bias correction term always exists (see Supplement B.2.2). Laplacian perturbations are however preferred as the required noise scale can be expressed analytically without additional optimisation (Balle and Wang, 2018), and as they give stricter privacy guarantees with $\delta = 0$.

Alternatively, unbiased importance weighted estimates can be computed directly by noising the weights instead of the coefficients of the logistic regression. While this procedure removes the bias of the estimates and can also be shown to be consistent, it increases the variance to a greater extent than noising the coefficients does, and is thus only sustainable when small amounts of data are released. Please refer to Supplement A.1 for more details.

### 3.4 PRIVATISING NEURAL NETWORKS

If logistic regression fails to give accurate density ratio estimates, for example because of biases introduced by the classifier's linearity assumptions, a more complex discriminator in the form of a neural network can be trained. We can train DP classification neural networks for the aim of likelihood ratio estimation with stochastic gradient decent (SGD) by clipping the gradients and adding calibrated Gaussian noise at each step of the SGD, see e.g. Abadi et al. (2016). The noised gradients are then added up in a *lot* before the descent step where lots resemble mini-batches.

These optimisation algorithms are commonly formulated for the case when the complete dataset is private. However, in our setting, $N_D$ observations are private and $N_G$ observations are non-private. Thus, we can define a relaxed version of DP SGD. Algorithm 1 provides an overview of our proposed method. We highlight the modifications to Algorithm 1 from Abadi et al. (2016) in blue.

**Proposition 3.** *Each step in the SGD outlined in Algorithm 1 is $(\epsilon, \delta)$-differentially private w.r.t the lot and $(\mathcal{O}(q\epsilon), \delta)$ differentially private w.r.t the full dataset where $q = \frac{L}{N_D+N_G}$ and $\sigma = \sqrt{2\log\left(\frac{1.25}{\delta}\right)}/\epsilon$.*

The differential privacy w.r.t a lot follows directly from the observation that the gradients of the synthetic data are already private. Further, the labels of the synthetic data are public knowledge. Lastly, the differential privacy w.r.t the dataset follows from the amplification theorem (Kasiviswanathan et al., 2011), the fact that sampling one particular private observation within a lot of size $L$ is $q = \frac{L}{N_D+N_G}$, and the reasoning behind the moment accountant of Abadi et al. (2016). We still clip the gradients of the public dataset

---

**Algorithm 1:** Relaxed DP SGD

**Input:** Examples $x_{1:N_D}, y_{1:N_D}$ from the DGP and $x_{N_D+1:N_D+N_G}, y_{N_D+1:N_D+N_G}$ from the SDGP, loss function $\mathcal{L}(\theta) = \frac{1}{N_G+N_D}\sum_i \mathcal{L}(\theta, x_i, y_i)$. Parameters: learning rate $\eta_t$, noise scale $\sigma$, expected lot size $L$, gradient norm bound $C$.

1 **Initialise** $\theta_0$ randomly
2 **for** $t \in [T]$ **do**
3    Construct a random subset $L_t \subset \{1, \dots, N_D + N_G\}$ by including each index independently at random with probability $\frac{L}{N_D+N_G}$
4    **Compute gradient**
5    For each $i \in L_t$, compute $g_t(x_i, y_i) \leftarrow \Delta_{\theta_t}\mathcal{L}(\theta_t, x_i, y_i)$
6    **Clip gradient**
7    $\bar{g}_t(x_i, y_i) \leftarrow g_t(x_i, y_i)/\max(1, \frac{||g_t(x_i,y_i)||_2}{C})$
8    **Add noise**
9    $\tilde{g}_t \leftarrow \frac{1}{L}\sum_{i \in L_t}(\bar{g}_t(x_i, y_i) + N(0, \sigma^2 C^2 \mathbf{I})\mathbb{1}_{(y_i=1)})$, where $\mathbb{1}_{(y_i=1)}$ is 1 if $y_i = 1$ and 0 otherwise
10    **Descent**
11    $\theta_{t+1} \leftarrow \theta_t + \eta_t \tilde{g}_t$

**Output:** $\theta_T$ and the overall privacy cost $(\epsilon, \delta)$ using the moment's accountant of Abadi et al. (2016) with sampling probability $q = \frac{L}{N_D+N_G}$.

---

as their influence will otherwise be overproportional under strong maximum norm assumptions.

### 3.5 GAN DISCRIMINATOR WEIGHTS

The downside of the aforementioned likelihood ratio estimators (Equation (4), Equation (5), Algorithm 1) is that their training requires an additional privacy budget which has to be added to the privacy budget used to learn the SDGP. If we however use a GAN such as DPGAN or PATE-GAN for private synthetic data generation, we can use the GAN's discriminator for the computation of the importance weights. According to the post processing theorem, these importance weights can be released without requiring an additional privacy budget. In contrast to the weights computed from DP classification networks, this approach is more robust and requires less hyperparameter tuning (confer to Section 4).

## 4 EXPERIMENTS

We demonstrate the benefits of using debiased IW for DP data release with a large-scale experimental study comparing three different SDGPs (DPGAN, DPCGAN, PrivBayes) on six real-world data sets (Iris, TGFB, Boston, Breast, Banknote, MNIST) for two different privacy budgets, $\epsilon \in \{1, 6\}$. We stress that debiasing comes with

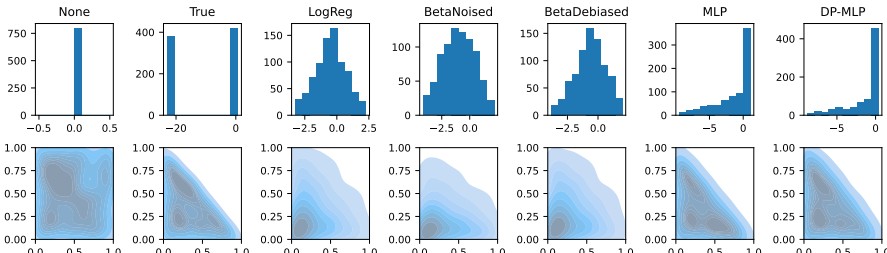

Figure 1: Kernel density plots of 100 observations sampled from a two dimensional uniform square distribution as SDGP (bottom left) and a uniform triangle distribution as DGP (second figure in second row). The first row depicts histograms of the computed weights starting with the true importance weights (True). The DP weights were privatised with $\epsilon = 1$, and the regularisation was chosen as $\lambda = 0.1$. The second row illustrates the importance weighted synthetic observations. We observe that while BetaDebiased corrects the weights of the logistic regression, the complex nature of the MLPs allows a better modelling of the DGP even in this simple setting.

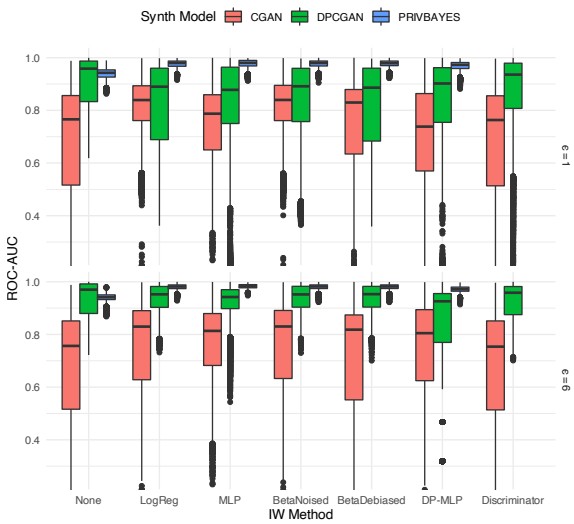

Figure 2: ROC-AUC score distributions calculated via chains of parameters sampled from a Bayesian logistic regression model fit on synthesised Banknote data across 10 seeds.

little overhead to the actual computations. As we see in Supplement C.2, the computations of the logistic regression and neural network importance weight estimates take less than one and a half minutes to train, even on MNIST. These weight estimators can be applied to any kind of synthetic data generation model, while the importance weights of the GAN discriminator can be computed in a single line of Python code and do not require any additional concerns regarding the privacy budget. Please see `https://github.com/sghalebikesabi/importance-weighted-differential-privacy` for the implementation.

**Computation of importance weights** After fitting the SDGP on the scaled true data, we weight each synthetic

observation with importance weights. Based on the train and the synthetic data, we apply one of the following IW approaches: weights computed from a non-private logistic regression (LogReg), its DP alternative introduced by Ji and Elkan (2013) (BetaNoised), or our debiased proposal (BetaDebiased), and likelihood ratios estimated by a non-private multi-layer perceptron (MLP), or a DP-MLP trained using Algorithm 1. We also compare to the naive estimator using uniform weights without IW (called 'None').

Please refer to Supplement C.1 for more details on the implementation and the hyperparameters used in our experiments. In Supplement C.8, we provide a comparison to the experimental results reported by related papers. Because of the large scale of our experimental study, we present only the most important results in this section, and give a complete overview in Supplement C. The code and data for all experiments can be found in the Supplements, and will be made available online.

### 4.1 TOY EXAMPLE

We start our analysis with a simple example to illustrate the benefits of the different weighting schemes. We assume that the synthetic data is sampled from a two-dimensional uniform distribution from 0 to 1 whereas the true data follows a uniform distribution on the lower triangle given by $x_1 + x_2 < 1$ for $x_1, x_2 \in [0, 1]$. This illustrative toy example was chosen for a fairer comparison of the logistic regression and the neural network based approaches. As we see in Figure 1, the weighted kernel density estimate (KDE) of BetaDebiased is closer to the LogReg weighted KDE, and also the true KDE compared to the BetaNoised KDE.

### 4.2 UCI DATA SETS

**Datasets and preprocessing** We performed additional experiments on four UCI datasets of different characteristics

| | IW | Breast | | | Banknote | | |
|---|---|---|---|---|---|---|---|
| | | DPGAN | DPCGAN | PrivBayes | DPGAN | DPCGAN | PrivBayes |
| WST→ | None | $2.3665_{\pm 0.0982}$ | $1.5853_{\pm 0.1333}$ | $2.1117_{\pm 0.1740}$ | $0.4746_{\pm 0.0214}$ | $0.7442_{\pm 0.0333}$ | $0.3237_{\pm 0.0162}$ |
| | BetaNoised | $1.4337_{\pm 0.1114}$ | $2.2232_{\pm 0.2325}$ | $1.2322_{\pm 0.0823}$ | $0.2509_{\pm 0.0436}$ | $0.4355_{\pm 0.0456}$ | $0.2318_{\pm 0.0035}$ |
| | BetaDebiased | $1.8922_{\pm 0.1237}$ | $1.9913_{\pm 0.3507}$ | $\mathbf{1.1825_{\pm 0.0933}}$ | $0.4015_{\pm 0.0766}$ | $0.4618_{\pm 0.0832}$ | $0.2369_{\pm 0.0061}$ |
| | DP-MLP | $1.4570_{\pm 0.1492}$ | $1.0315_{\pm 0.1415}$ | $1.2190_{\pm 0.0795}$ | $\mathbf{0.2035_{\pm 0.0427}}$ | $0.4298_{\pm 0.0433}$ | $\mathbf{0.0456_{\pm 0.0061}}$ |
| | Discriminator | $\mathbf{1.0007_{\pm 0.0004}}$ | $\mathbf{1.0001_{\pm 0.0001}}$ | - | $0.3382_{\pm 0.0399}$ | $\mathbf{0.1087_{\pm 0.0415}}$ | - |
| | LogReg | $1.6451_{\pm 0.1168}$ | $2.2953_{\pm 0.2121}$ | $1.4663_{\pm 0.1152}$ | $0.2508_{\pm 0.0432}$ | $0.4348_{\pm 0.0460}$ | $0.2348_{\pm 0.0034}$ |
| | MLP | $1.6129_{\pm 0.1404}$ | $1.0709_{\pm 0.1579}$ | $1.4141_{\pm 0.1216}$ | $0.0913_{\pm 0.0259}$ | $0.3860_{\pm 0.0452}$ | $0.0021_{\pm 0.0004}$ |
| $\beta$ MSE↓ | None | $2.0643_{\pm 0.2012}$ | $4.9828_{\pm 1.5701}$ | $2.3904_{\pm 0.1050}$ | $11.0215_{\pm 1.8377}$ | $19.3243_{\pm 3.7708}$ | $8.1724_{\pm 0.3987}$ |
| | BetaNoised | $2.7532_{\pm 0.2650}$ | $2.5025_{\pm 0.3763}$ | $2.1144_{\pm 0.2400}$ | $8.4298_{\pm 1.0383}$ | $15.2862_{\pm 4.0365}$ | $5.7001_{\pm 0.1885}$ |
| | BetaDebiased | $2.8337_{\pm 0.3842}$ | $\mathbf{2.2324_{\pm 1.0446}}$ | $\mathbf{1.8266_{\pm 0.2392}}$ | $\mathbf{8.3508_{\pm 2.3127}}$ | $12.9909_{\pm 5.9024}$ | $6.6862_{\pm 0.1458}$ |
| | DP-MLP | $2.3965_{\pm 0.2083}$ | $3.8865_{\pm 0.6043}$ | $2.3130_{\pm 0.2195}$ | $17.1597_{\pm 2.5448}$ | $16.4618_{\pm 4.1011}$ | $\mathbf{3.5519_{\pm 0.2895}}$ |
| | Discriminator | $\mathbf{1.4591_{\pm 0.1837}}$ | $4.0612_{\pm 0.9523}$ | - | $12.5471_{\pm 2.3124}$ | $\mathbf{10.9282_{\pm 5.4283}}$ | - |
| | LogReg | $2.6934_{\pm 0.2667}$ | $2.2156_{\pm 0.3366}$ | $1.5333_{\pm 0.2138}$ | $8.4760_{\pm 1.0406}$ | $15.2964_{\pm 4.0396}$ | $5.6751_{\pm 0.1785}$ |
| | MLP | $2.3999_{\pm 0.2040}$ | $3.8343_{\pm 0.7032}$ | $1.6581_{\pm 0.2020}$ | $17.9390_{\pm 2.4926}$ | $15.5211_{\pm 4.2147}$ | $2.6286_{\pm 0.3761}$ |
| MLP ROC-AUC← | None | $0.6374_{\pm 0.0421}$ | $0.6791_{\pm 0.0966}$ | $0.8366_{\pm 0.0579}$ | $0.8546_{\pm 0.0213}$ | $0.6863_{\pm 0.0436}$ | $0.7630_{\pm 0.0495}$ |
| | BetaNoised | $0.6110_{\pm 0.0477}$ | $0.6546_{\pm 0.0727}$ | $0.7076_{\pm 0.0983}$ | $0.8495_{\pm 0.0274}$ | $0.6063_{\pm 0.0510}$ | $0.8943_{\pm 0.0173}$ |
| | BetaDebiased | $0.6820_{\pm 0.0510}$ | $0.7173_{\pm 0.0842}$ | $\mathbf{0.8557_{\pm 0.0765}}$ | $\mathbf{0.8729_{\pm 0.0310}}$ | $0.5868_{\pm 0.1005}$ | $0.7632_{\pm 0.0517}$ |
| | DP-MLP | $\mathbf{0.7942_{\pm 0.0404}}$ | $0.5686_{\pm 0.0823}$ | $0.7353_{\pm 0.0887}$ | $0.7697_{\pm 0.0419}$ | $0.5657_{\pm 0.0570}$ | $\mathbf{0.8953_{\pm 0.0299}}$ |
| | Discriminator | $0.6992_{\pm 0.0839}$ | $\mathbf{0.7290_{\pm 0.0720}}$ | - | $0.8695_{\pm 0.0167}$ | $\mathbf{0.7114_{\pm 0.0424}}$ | - |
| | LogReg | $0.6631_{\pm 0.0469}$ | $0.6484_{\pm 0.1081}$ | $0.7618_{\pm 0.1019}$ | $0.8172_{\pm 0.0327}$ | $0.6034_{\pm 0.0534}$ | $0.9102_{\pm 0.0129}$ |
| | MLP | $0.7730_{\pm 0.0412}$ | $0.7358_{\pm 0.1017}$ | $0.7573_{\pm 0.0738}$ | $0.8291_{\pm 0.0333}$ | $0.5974_{\pm 0.0627}$ | $0.8594_{\pm 0.0231}$ |

Table 1: Mean and standard error over 10 runs for ($\epsilon = 1$, $\delta = N_D^{-1} - e^{-6}$) on the Breast and Banknote data. Best score out of the private methods is marked in bold.

as decribed in Supplement C.1: Iris, Banknote, Boston, and Breast. Similarly to Chaudhuri et al. (2011); Ji and Elkan (2013), we scale all data to a feature range from 0 to 1. We use a train-test split of 80%. In all experiments we fix $\delta$ to $N_D^{-1} - 10^{-6}$, and choose $\epsilon \in \{1, 6\}$. We refer to Supplement C.7 for a complete overview of the results.

**Synthetic data generators** We used DPCGAN (Torkzadehmahani et al., 2019), DPGAN (Xie et al., 2018), and their corresponding non-DP analogues (CGAN and CGAN) to generate DP synthetic data of the same size as the training data set. Additionally we also consider PrivBayes (Zhang et al., 2017), a DP Bayesian Network, as a potential SDGP.

**Hyperparameter tuning** Note that hyperparameter tuning is essentially non-private, and has to be accounted for in the privacy budget. Since hyperparemeter tuning in a DP setting is an unresolved problem (Liu and Talwar, 2019; Rosenblatt et al., 2020; Papernot and Steinke, 2021), we follow Jordon et al. (2019) and tune the hyperparameters of the underlying baselines on private validation data sets. However, we propose default parameters for our methods. This leads to an over-optimistic presentation of the baseline performance, and a conservative presentation of our extensions.

**Evaluation metrics** In order to show that IW decreases statistical bias, we train a linear prediction model on the synthetic data and approximate its bias. Since the true DGP is not known, we train the same linear predictor on the test data and report the mean squared error (MSE) between the test parameters and the parameters estimated on the SDGP, as $\beta$ MSE. We further analyse the divergence of the weighted SDGP and the DGP in a similar way by computing the Wasserstein (WST) distance w.r.t the test data. As one exemplary supervised downstream task, we consider the training of a linear downstream classifier or regressor on the synthetic data. This downstream predictor is then assessed by the error measured in the parameter vector compared to the parameters learnt using the test set ($beta$ MSE). As another downstream task, we train a one-hidden-layer MLP on the training data, and report the test prediction error as MLP ROC-AUC for classification tasks, and MLP MSE for regression tasks.

**Choice of budget split** We only present results for $\epsilon = 1$ in this section, and refer the reader to Supplement C.7 for further results with $\epsilon = 6$. If the weight computation procedure requires a separate privacy budget (e.g. if the weights are computed by a separate MLP or logistic regression), we spend 10% of the $\epsilon$-budget on fitting the SDGP and 30% of the $\delta$-budget on the weight computation; the complete budget can be spent on fitting the SDGP if no weights, or the weights of the discriminator are used. In Supplement C.3, we evaluate a range of different privacy splits on the Breast and Boston data.

**Results** In Tables 1 and 2, we see that the performance of the models mostly improved when weighted with any type of estimated weights. Although the best inference for

| | IW | DPGAN | PrivBayes |
|---|---|---|---|
| **WST →** | None | $2.2013_{\pm 0.0945}$ | $1.3938_{\pm 0.0231}$ |
| | BetaNoised | $2.0922_{\pm 0.0419}$ | $1.3009_{\pm 0.0338}$ |
| | BetaDebiased | $2.0930_{\pm 0.0393}$ | $1.2705_{\pm 0.0290}$ |
| | DP-MLP | $2.0542_{\pm 0.0184}$ | $\mathbf{1.0265_{\pm 0.0035}}$ |
| | Discriminator | $\mathbf{2.0145_{\pm 0.0141}}$ | - |
| | LogReg | $2.2051_{\pm 0.0819}$ | $1.4078_{\pm 0.0492}$ |
| | MLP | $2.0350_{\pm 0.0158}$ | $1.0072_{\pm 0.0009}$ |
| **β MSE ↓** | None | $0.1867_{\pm 0.0434}$ | $\mathbf{0.0011_{\pm 0.0002}}$ |
| | BetaNoised | $0.1761_{\pm 0.0948}$ | $0.0088_{\pm 0.0028}$ |
| | BetaDebiased | $\mathbf{0.0667_{\pm 0.0188}}$ | $0.0077_{\pm 0.0022}$ |
| | DP-MLP | $0.1530_{\pm 0.0812}$ | $0.0048_{\pm 0.0024}$ |
| | Discriminator | $0.1567_{\pm 0.1825}$ | - |
| | LogReg | $0.0749_{\pm 0.0279}$ | $0.0037_{\pm 0.0016}$ |
| | MLP | $0.1476_{\pm 0.0804}$ | $0.0008_{\pm 0.0002}$ |
| **MLP MSE →** | None | $1.8851_{\pm 0.5262}$ | $0.1973_{\pm 0.0108}$ |
| | BetaNoised | $1.0057_{\pm 0.1973}$ | $0.2200_{\pm 0.0154}$ |
| | BetaDebiased | $\mathbf{0.9024_{\pm 0.1244}}$ | $0.2139_{\pm 0.0122}$ |
| | DP-MLP | $0.9462_{\pm 0.1702}$ | $\mathbf{0.1877_{\pm 0.0174}}$ |
| | Discriminator | $1.6256_{\pm 0.2394}$ | - |
| | LogReg | $1.0606_{\pm 0.2648}$ | $0.2515_{\pm 0.0305}$ |
| | MLP | $1.0979_{\pm 0.2225}$ | $0.1697_{\pm 0.0079}$ |

Table 2: Mean and standard error over 10 runs for ($\epsilon = 1$, $\delta = N_D^{-1} - e^{-6}$) on the Boston Housing data. Best score out of the private methods is marked in bold.

| IW | $\beta$ MSE ↓ | MLP ROC-AUC ↑ |
|---|---|---|
| None | $0.6605_{\pm 0.0384}$ | $0.8502_{\pm 0.0386}$ |
| BetaNoised | $0.6247_{\pm 0.0184}$ | $0.8766_{\pm 0.0086}$ |
| BetaDebiased | $0.6240_{\pm 0.0179}$ | $\mathbf{0.8783_{\pm 0.0093}}$ |
| DP-MLP | $\mathbf{0.5813_{\pm 0.0246}}$ | $0.8683_{\pm 0.0055}$ |
| Discriminator | $0.6242_{\pm 0.0140}$ | $0.8631_{\pm 0.0310}$ |
| LogReg | $0.6234_{\pm 0.0183}$ | $0.8770_{\pm 0.0092}$ |
| MLP | $0.5707_{\pm 0.0207}$ | $0.8737_{\pm 0.0058}$ |

Table 3: Mean and standard error over 10 runs with standard errors for ($\epsilon = 9.64, \delta = 60,000^{-1} - e^{-6}$) on MNIST.

each data set is nearly always achieved after importance weighting, we notice that there are some rare cases where no importance weighting performs (insignificantly) better. For instance, we observe that the SDGP obtained with PrivBayes seems to be close to the true DGP of the Boston Housing data, and that importance weighting is no longer helpful. In settings where the SDGP and the DGP are really close, it is possible that the effects of additional variance induced by estimating and privatising the importance weights (where appropriate) cancels out the reduction in bias. This effect might be mitigated with hyperparameter tuning. Further, we note that debiasing the logistic regression weights mainly results in better performance. Even though we experience a slight drop in performance from BetaNoised to BetaDebiased in some rare cases, this can be explained by randomness in the data set as we show in Supplement Table 6 that the weights estimated by BetaDebiased are significantly closer to the

true LogReg weights than the importance weights given by BetaNoised. If a GAN is used as SDGP, and the data curator is hesitant to release additional importance weights, the discriminator weights nearly always lead to an improvement in results without requiring additional computations. To further illustrate the practical meaning of debiasing, we have included an exemplary case study in Supplement C.6.

### 4.3 BAYESIAN UPDATING WITH IW

We investigate the effectiveness of IW in a Bayesian learning setting as per Equation 3. We evaluated and compared the performance of these weighted posteriors alongside the standard non-weighted posterior by applying them to learning the parameters of models for various regression tasks. Figure 2 shows the ROC-AUC scores associated with the Bayesian predictive distribution arising from integration over the posterior of a Bayesian logistic regression model fit on synthesised versions of the Banknote dataset. We observe that the ROC-AUC under PrivBayes' synthetic data is significantly improved upon across all IW methods, with similar gains made to the median performance under CGAN's synthetic data. Additionally, most of the methods help in decreasing variability in the results, especially DP-MLP and MLP. See Supplement C.5 for a full specification of the experimental details and for further results from fitting Bayesian linear regression and multinomial logistic regression models on the TGFB and Iris datasets respectively.

### 4.4 MNIST

Additionally, we assessed how IW performs in a high-dimensional setting such as a classification task on the MNIST dataset. Since PrivBayes does not scale to large data sets, we only evaluate DPCGAN as possible SDGP. For this we follow the setup by Torkzadehmahani et al. (2019) for $\epsilon = 9.64$ and $\delta = 6000^{-1} - 10^{-6}$. We observe in Table 3 that all IW methods improve upon the state of the art.

## 5 DISCUSSION

In this paper, we investigated importance weighting methods to correct for biases in downstream estimation tasks when using differentially private synthetic data. While classification algorithms can be used to estimate the required importance weights, noise must be added in order to maintain privacy. We presented methods to debias inference based on privatised weights estimated by logistic regression, developed private estimation procedures allowing the complexity of neural networks to be leveraged for weight estimation, and proposed using inbuilt discriminator weights from GAN synthetic data generation to avoid increases to the privacy budget.

Following these developments, we advocate that future releases of DP synthetic data are augmented with privatised importance weights to allow researchers to conduct unbiased downstream model estimation. Future work will focus on improved hyperparameter tuning practises to choose the optimal IW approach for the task and dataset at hand.

## Acknowledgements

SG is a student of the EPSRC CDT in Modern Statistics and Statistical Machine Learning (EP/S023151/1) and receives funding from the Oxford Radcliffe Scholarship and Novartis. HW is supported by the Feuer International Scholarship in Artificial Intelligence. JJ was funded by the Ayudas Fundación BBVA a Equipos de Investigación Cientifica 2017 and Government of Spain's Plan Nacional PGC2018-101643-B-I00 grants whilst working on this project. SJV is supported by the University of Warwick, University of Warwick and German Resarch Centre for Aritifical Intelligence. CH is supported by The Alan Turing Institute, Health Data Research UK, the Medical Research Council UK, the EPSRC through the Bayes4Health programme Grant EP/R018561/1, and AI for Science and Government UK Research and Innovation (UKRI).

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

# A ADDITIONAL MATERIAL

## A.1 UNBIASED IMPORTANCE WEIGHTING BY OUTPUT PERTURBATION

A simple approach to ensure DP of an algorithm is to add noise (Dwork et al., 2006) to its output, that is the estimated importance weights of the synthetic data. We establish general results under which such a noise perturbation of an unbiased non-private weights algorithm $\widehat{w}(x)$ preserves the unbiasedness of IS estimation.

**Theorem 2.** *Let $\sigma^2(h)/N$ denote the variance of the IS estimate $I_N(h|w)$ defined in Equation (2). Then the IS estimator $I_N(h|w^*)$ using noise perturbed importance weights $w^*(x_i) = \widehat{w}(x_i) + \zeta_i$, where $\zeta_i$ are i.i.d. and $\mathbb{E}[\exp(\zeta_i)] = 1$, is unbiased and has variance $\sigma^{*2}(h)/N$ where*

$$\sigma^{*2}(h) = \sigma^2(h) + \text{Var}\left[\exp(\zeta)\right] \mathbb{E}_{p_G}[(\widehat{w}(x)h(x))^2]. \tag{6}$$

We refer the reader to Supplement B.3 for the proof. In the following we will analyse how the noise $\zeta$ has to be chosen to ensure DP.

**Corollary 1.** *The IS estimator with importance weights defined by*

$$\log w^*(x_i) = \widehat{\beta}^T x_i + \zeta_i \tag{7}$$

$$\text{for} \quad \zeta_i \sim Laplace(\log(1-\rho^2), \rho) \quad \text{and} \quad \rho = \frac{2\sqrt{d}}{N_D\lambda\epsilon} < 1$$

*is $(N_S\epsilon, 0)$-differentially private. It is further unbiased and for $\rho < \frac{1}{2}$ has variance as defined in equation 6:*

$$\text{Var}\left[\exp(\zeta)\right] = \exp(2\log(1-\rho^2)) \left(\frac{1}{1-4\lambda^2} - \frac{1}{(1-\lambda^2)^2}\right).$$

Note that privacy budget is additive. If we want to release $N_S$ DP weights, we thus have to scale the noise proportional to $N_S$. Although this approach increases the variance of the estimator, it remains unbiased.

A limitation of this approach is that $\rho < \frac{1}{2}$. Alternatively, Blum et al. (2005) show that adding Gaussian noise $\zeta' \sim N(0, \frac{2}{\epsilon^2}S(f)^2 \log\frac{2}{\delta})$ to an algorithm $f$ ensures $(\epsilon, \delta)$-DP for $\delta > 0$. From our analysis it follows that we could adjust Corollary 1 as follows.

**Corollary 2.** *The IS estimator with importance weights defined by*

$$\log w^*(x_i) = \widehat{\beta}^T x_i + \zeta_i'$$

$$\text{for} \quad \zeta_i' \sim N(-\frac{\gamma^2}{2}, \gamma^2) \quad \text{and} \quad \gamma = \sqrt{\frac{8d}{(N_D\lambda\epsilon)^2}\log\frac{2}{\delta}}$$

*is $(N_S\epsilon, \delta)$-differentially private with $\delta > 0$ and $\epsilon < 1$. It is further unbiased and has variance as defined in equation 6 with $\text{Var}\left[\exp(\zeta')\right] = \gamma^2$.*

This result trivially extends to the case of $\epsilon \geq 1$ with accordingly adjusted noise scales following results from Balle and Wang (2018).

**Sources of Bias and Variance.** This analysis gives us insights on two sources of bias and variance. The first one is the bias and/or variance introduced by *privatising* the weights. The estimator of Ji and Elkan (2013) is biased but as a result adds noise with a smaller variance, whereas to be unbiased by noising the weights we have to pay a price of increasing the variance, e.g., by adding more noise or by releasing fewer samples. The second source is the bias and variance introduced by *estimating* the weights through the classifier. The importance weighting procedure is only unbiased when we know exactly how to estimate the true weights. Using a logistic regression to estimate these cannot reasonably be considered as unbiased for any complicated data. However, using an arbitrarily complex classifier such as a classification neural network could arguably be considered as less biased at estimating the density ratio if it converges, but possibly increases the variance of the estimators due to the increased number of parameters to learn.

## A.2 POST-PROCESSING OF LIKELIHOOD RATIOS

The performance of importance weighting can suffer from a heavy right tailed distribution of the likelihood ratio estimates which increases the variance of downstream estimators. A simple remedy is tempering: for a $\tau \in [0,1]$ the weights $\{\widehat{w}(x_i)^\tau\}_{i \in \{1,\dots,N_G\}}$ are less extreme.

Alternatively, Vehtari et al. (2015) propose Pareto smoothed IS (PSIS). This procedure requires to fit a generalised Pareto distribution to the upper tail of the distribution of the simulated importance ratios. Their algorithm does not only post-hoc stabilise IS, but also reports a warning when the estimated shape parameter of the Pareto distribution exceeds a certain threshold. Similarly, Koopman et al. (2009) propose a test to detect whether importance weights have finite variance. In both warnings, there are certain characteristics of the DGP which are not captured by the SDGP and the resulting IS estimates are likely to be unstable. This warning can thus be understood as a general indicator for unsuitable proposal distributions. For large shape parameters the data owner should not release the SDGP. It is also computationally more efficient than comparable distribution divergences such as maximum mean discrepancy or Wasserstein distance. We must also consider that unlike traditional IS where the importance weights are known (at least up to normalisation), here they are being estimated from data, providing further motivation for regularisation.

Aside from unstable likelihood ratios, the computed importance weights can suffer from the inability of the classification method to correctly capture the density ratios. To mitigate this problematic, Turner et al. (2019) propose post-calibration of the likelihood ratios in a non-private setting. If we can assume that the data analyst has access to a small dataset of the DGP, as e.g. in Wilde et al. (2020), we can make use of post-calibration methods, such as beta calibration (Kull et al., 2017).

# B PROOFS

## B.1 PROPOSITION 1: BIAS AND VARIANCE OF ALGORITHM 1 OF JI & ELKAN (2013)

Consider Ji and Elkan (2013) Algorithm 1, where under the assumption that $\frac{p(y=1)}{p(y=0)} \approx \frac{N_D}{N_S} = 1$, the unprivatised importance weights are estimated using logistic regression

$$\widehat{w}(x_i) = \frac{p^*(y=1|x_i)}{p^*(y=0|x_i)} = \exp\left(\widehat{\beta}^T x_i\right),$$

and then the privacy preserving process adds noise to the $\widehat{\beta}$ coefficients of this logistic regression $\beta^* = \widehat{\beta} + \zeta$ with $\zeta \sim \text{Laplace}(2\sqrt{d}/(N_D \lambda \epsilon))$, a vector of length $d$, to generate privatised estimates of the importance weights

$$\overline{w}(x_i) = \exp\left(\beta^{*^T} x_i\right) = \exp\left(\widehat{\beta}^T x_i\right) \cdot \exp\left(\zeta x_i\right). \tag{8}$$

The following proposition proves that $\overline{w}(x_i)$ is a *biased* estimate of $\widehat{w}(x_i)$, the consequences being that if the 'true' importance weight really is given by a logistic regression then the procedure of Ji and Elkan (2013) will be biased.

**Proposition 1.** *Let $\overline{w}$ denote the importance weights computed by noise perturbing the regression coefficients as in Equation (8) (Ji and Elkan, 2013, Algorithm 1). The importance sampling estimator $I_N(h|\overline{w})$ is biased.*

**Proof.** Firstly, we show that $\overline{w}(x_i)$ is not an unbiased estimate of $\widehat{w}(x_i)$

$$\mathbb{E}_\zeta\left[\overline{w}(x_i)\right] = \mathbb{E}_\zeta\left[\exp\left(\widehat{\beta}^T x_i\right) \cdot \exp\left(\zeta x_i\right)\right]$$
$$= \mathbb{E}_\zeta\left[\widehat{w}(x_i) \cdot \exp\left(\zeta x_i\right)\right]$$
$$\neq \widehat{w}(x_i).$$

As a consequence, we show that even if the true density ratio can be captured by a logistic regression, i.e. there exists $\beta_0$ such that $\frac{p_D(x)}{p_G(x)} = \exp\left(\beta_0^T x\right)$, then the importance sampling estimator

$$I_N(h|\overline{w}) = \frac{1}{N} \sum_{i=1}^N \overline{w}(x_i) h(x_i), \quad x_i \sim p_G(\cdot),$$

with $\overline{w}(\cdot)$ calculated using 'privatised' $\beta^* = \beta_0 + \zeta$, $\zeta$ distributed as above, is a biased estimate of $\mathbb{E}_{p_D}[h(x)]$. Indeed, we have

$$\mathbb{E}_{x_{1:N} \sim p_G} \left[ \frac{1}{N} \sum_{i=1}^{N} \overline{w}(x_i) h(x_i) \right] = \mathbb{E}_{x_{1:N} \sim p_G} \left[ \frac{1}{N} \sum_{i=1}^{N} \exp\left(\beta_0^T x_i\right) \cdot \exp\left(\zeta x_i\right) h(x_i) \right]$$

$$= \frac{1}{N} \sum_{i=1}^{N} \mathbb{E}_{x_i \sim p_G} \left[ w\left(x_i\right) \cdot \exp\left(\zeta x_i\right) h(x_i) \right]$$

$$= \frac{1}{N} \sum_{i=1}^{N} \mathbb{E}_{x_i \sim p_D} \left[ \exp\left(\zeta x_i\right) h(x_i) \right]$$

$$\neq \mathbb{E}_{x_i \sim p_D} \left[ h(x_i) \right].$$

The proof of Proposition 1 provides several insights on what is required for an unbiased estimator. The fact that the bias depends explicitly on the observation suggests either 1) asking the data curator to debias the noise given the synthetic data they are about to release or 2) adding noise to the weights themselves rather to the process of how they are calculated.

Ji and Elkan (2013) compute the variance of the estimator $\beta^* = \widehat{\beta} + \zeta$ where $\zeta \sim \text{Laplace}(\frac{4(d+1)d}{(N_D \lambda \epsilon)^2})$ as

$$\text{Var}(\beta^*) = \text{Var}(\widehat{\beta}) + \text{Var}(\zeta) = \text{Var}(\widehat{\beta}) + \frac{4(d+1)d}{(N_D \lambda \epsilon)^2}.$$

They show that the asymptotic variance of importance sampling with the unperturbed weights obtained from the logistic regression $w_{logreg}$ can be upper bounded by

$$\text{Var}(I_N(h, w_{logreg})) = \alpha^T \text{Var}(\widehat{\beta}) \alpha = \alpha^T \frac{d I_d}{N_D \lambda^2} \alpha$$

with

$$\alpha = \frac{\sum_{x_i, x_j \in D} e^{\beta_0^T (x_i + x_j)} \left( h\left(x_i\right) - h\left(x_j\right) \right) \left(x_i - x_j\right)}{\sum_{x_i, x_j \in E} e^{\beta_0^T (x_i + x_j)}},$$

where $\beta_0$ optimises the loss function of a logistic regression on fixed $G$ and the true distribution of $D$. The asymptotic variance of the importance sampling estimator with the weights $w^*_{logreg}$ from the logistic regression with parameter $\beta^*$ is then

$$\text{Var}(I_N(h, w^*_{logreg})) = \alpha^T \text{Var}(\beta^*) \alpha = \alpha^T \left( \frac{d I_d}{N_D \lambda^2} + \frac{4(d+1)d}{(N_D \lambda \epsilon)^2} \right) \alpha.$$

## B.2 PROPOSITION 2: DEBIASING OF JI & ELKAN (2013)

As prescribed by Ji and Elkan (2013) Algorithm 1, consider importance weights

$$\overline{w}(x_i) = \exp\left(\beta^{*T} x_i\right) = \exp\left(\widehat{\beta}^T x_i\right) \cdot \exp\left(\zeta^T x_i\right). \tag{9}$$

for privacy preserved $\widehat{\beta}$ coefficients of this logistic regression $\beta^* = \widehat{\beta} + \zeta$ with $\zeta \sim \text{Laplace}(2\sqrt{d}/(N_D \lambda \epsilon))$, a vector of length $d$. Proposition 1 proved that using $\overline{w}(\cdot)$ resulted in biased expectation estimation. However, Proposition 2 demonstrates that we can debias this in closed form.

**Proposition 2.** *Let $\overline{w}$ denote the importance weights computed by noise perturbing the regression coefficients as in Equation (9) (Ji and Elkan, 2013, Algorithm 1) with $\zeta \sim p_\zeta$. Define*

$$b(x_i) := 1 / \mathbb{E}_{\zeta \sim p_\zeta}[\exp\left(\zeta^T x_i\right)],$$

*and adjusted importance weight*

$$\overline{w}^*(x_i) = \overline{w}(x_i) \cdot b(x_i) = \widehat{w}(x_i) \cdot \exp\left(\zeta^T x_i\right) \cdot b(x_i).$$

*The importance sampling estimator $I_N(h|\overline{w}^*)$ is unbiased and $(\epsilon, 0)$-differentially private. The variance of estimator $I_N(h|\overline{w}^*)$ has the following decomposition*

$$\text{Var}_{p_G^*}\left[I_N(h|\overline{w}^*)\right] = \frac{\overline{\sigma}^{*2}(h)}{N} + \left(1 - \frac{1}{N}\right)\overline{c}^*(h).$$

*with*

$$\begin{aligned}
\overline{\sigma}^{*2}(h) &= \sigma^2(h) + \mathbb{E}_{x\sim p_G}\left[h(x)^2\widehat{w}(x)^2\text{Var}_{\zeta\sim p_\zeta}\left[b(x)\exp(\zeta x)\right]\right], \\
\sigma^2(h) &= \text{Var}_{x\sim p_G}\left[h(x)\widehat{w}(x)\right], \\
\overline{c}^*(h) &= \mathbb{E}_{x,x'\sim p_G}\left[h(x)\widehat{w}(x)h(x')\widehat{w}(x')\left(\frac{b(x)b(x')}{b(x+x')} - 1\right)\right].
\end{aligned} \tag{10}$$

**Proof.** Consider $(x_1, \ldots, x_N, \zeta) \overset{i.i.d}{\sim} p_G^*$, i.e. $x_i \overset{i.i.d}{\sim} p_G$, $i = 1, \ldots, N$ and $\zeta \sim p_\zeta$ and

$$I_N(h|\overline{w}^*) = \frac{1}{N}\sum_{i=1}^{N} h(x_i)\widehat{w}(x_i)\exp\left(\zeta^T x_i\right) b(x_i),$$

then

$$\begin{aligned}
\mathbb{E}_{p_G^*}\left[I_N(h|\overline{w}^*)\right] &= \mathbb{E}_{x\sim p_G(x)}\mathbb{E}_{\zeta\sim p_\zeta}[h(x)\widehat{w}(x)\exp\left(\zeta^T x\right) b(x)] \\
&= \mathbb{E}_{x\sim p_G(x)}[h(x)\widehat{w}(x)b(x)\mathbb{E}_{\zeta\sim p_\zeta}[\exp\left(\zeta^T x\right)]] \\
&= \mathbb{E}_{x\sim p_G(x)}[h(x)\widehat{w}(x)] \\
&= \mathbb{E}_{x\sim p_D(x)}[h(x)]
\end{aligned}$$

and as a result $I_N(h|\overline{w}^*)$ is an unbiased estimator of $\mathbb{E}_{x\sim p_D(x)}[h(x)]$. The variance of estimator $I_N(h|\overline{w}^*)$ is given by

$$\begin{aligned}
\text{Var}_{p_G^*}\left[I_N(h|\overline{w}^*)\right] &= \frac{1}{N^2}\sum_{i=1}^{N}\text{Var}_{p_G^*}\left[h(x_i)\overline{w}^*(x_i)\right] + \frac{2}{N^2}\sum_{i=1}^{N}\sum_{j<i}\text{Cov}_{p_G^*}\left[h(x_i)\overline{w}^*(x_i), h(x_j)\overline{w}^*(x_j)\right] \\
&= \frac{\overline{\sigma}^{*2}(h)}{N} + \left(1 - \frac{1}{N}\right)\overline{c}^*(h).
\end{aligned} \tag{11}$$

where the weights are dependent under $p_G^*$ because $\zeta$ **is not** sampled independently for each $x_i$, it is only sampled once. The terms making up (11) are

$$\begin{aligned}
\overline{\sigma}^{*2}(h) &= \text{Var}_{p_G^*}\left[h(x)\widehat{w}(x)\exp(\zeta x)b(x)\right] \\
&= \mathbb{E}_{p_G^*}\left[(h(x)\widehat{w}(x)\exp(\zeta x)b(x))^2\right] - \mathbb{E}_{p_G^*}\left[h(x)\widehat{w}(x)\exp(\zeta x)b(x)\right]^2 \\
&= \mathbb{E}_{x\sim p_G}\left[h(x)^2\widehat{w}(x)^2\mathbb{E}_{\zeta\sim p_\zeta}\left[b(x)^2\exp(\zeta x)^2\right]\right] - \mathbb{E}_{p_G}\left[h(x)\widehat{w}(x)\right]^2 \\
&= \mathbb{E}_{x\sim p_G}\left[h(x)^2\widehat{w}(x)^2\left(\text{Var}_{\zeta\sim p_\zeta}\left[b(x)\exp(\zeta x)\right] + 1\right)\right] - \mathbb{E}_{p_G}\left[h(x)\widehat{w}(x)\right]^2 \\
&= \sigma^2(h) + \mathbb{E}_{x\sim p_G}\left[h(x)^2\widehat{w}(x)^2\text{Var}_{\zeta\sim p_\zeta}\left[b(x)\exp(\zeta x)\right]\right],
\end{aligned}$$

with $\mathbb{E}_{\zeta\sim p_\zeta}\left[b(x)\exp(\zeta x)\right] = 1$ by construction and $\sigma^2(h)$ defined in (10), and

$$\begin{aligned}
\overline{c}^*(h) &= \text{Cov}_{p_G^*}\left[h(x)\widehat{w}(x)\exp\left(\zeta^T x\right) b(x), h(x')\widehat{w}(x')\exp\left(\zeta^T x'\right) b(x')\right] \\
&= \mathbb{E}_{x,x'\sim p_G, \zeta\sim p_\zeta}\left[h(x)\widehat{w}(x)\exp\left(\zeta^T x\right) b(x) \cdot h(x')\widehat{w}(x')\exp\left(\zeta^T x'\right) b(x')\right] \\
&\quad - \mathbb{E}_{x,\zeta\sim p_G^*}\left[h(x)\widehat{w}(x)\exp\left(\zeta^T x\right) b(x)\right] \cdot \mathbb{E}_{x',\zeta\sim p_G^*}\left[h(x')\widehat{w}(x')\exp\left(\zeta^T x'\right) b(x')\right].
\end{aligned}$$

By $\mathbb{E}_\zeta\left[\exp\left(\zeta^T x\right) b(x)\right] = 1$, and $x, x' \overset{iid}{\sim} p_G$ the second term simplifies to

$$\mathbb{E}_{x\sim p_G^*}\left[h(x)\widehat{w}(x)\exp\left(\zeta^T x\right) b(x)\right] \cdot \mathbb{E}_{x'\sim p_G^*}\left[h(x')\widehat{w}(x')\exp\left(\zeta^T x'\right) b(x')\right] = \mathbb{E}_{x\sim p_G}\left[h(x)\widehat{w}(x)\right]^2.$$

The first term can be simplified as

$$\mathbb{E}_{x,x'\sim p_G, \zeta\sim p_\zeta}\left[h(x)\widehat{w}(x)\exp\left(\zeta^T x\right)b(x)\cdot h(x')\widehat{w}(x')\exp\left(\zeta^T x'\right)b(x')\right]$$

$$= \mathbb{E}_{x,x'\sim p_G}\left[h(x)\widehat{w}(x)h(x')\widehat{w}(x')b(x)b(x')\mathbb{E}_{\zeta\sim p_\zeta}\left[\exp\left(\zeta^T(x+x')\right)\right]\right]$$

$$= \mathbb{E}_{x,x'\sim p_G}\left[h(x)\widehat{w}(x)h(x')\widehat{w}(x')\frac{b(x)b(x')}{b(x+x')}\right]$$

$$= \mathbb{E}_{x,x'\sim p_G}\left[h(x)\widehat{w}(x)h(x')\widehat{w}(x')\left(\frac{b(x)b(x')}{b(x+x')}-1\right)\right]$$

$$\quad + \mathbb{E}_{x\sim p_G}\left[h(x)\widehat{w}(x)\right]\mathbb{E}_{x'\sim p_G}\left[h(x')\widehat{w}(x')\right]\quad\text{(indep.)}$$

$$= \mathbb{E}_{x,x'\sim p_G}\left[h(x)\widehat{w}(x)h(x')\widehat{w}(x')\left(\frac{b(x)b(x')}{b(x+x')}-1\right)\right]$$

$$\quad + \mathbb{E}_{x\sim p_G}\left[h(x)\widehat{w}(x)\right]^2.$$

As a result

$$\bar{c}^*(h) = \mathbb{E}_{x,x'\sim p_G}\left[h(x)\widehat{w}(x)h(x')\widehat{w}(x')\left(\frac{b(x)b(x')}{b(x+x')}-1\right)\right]$$

### B.2.1 Special Case 1: Laplace Noise

Recall that $x_i$ and $\zeta$ are $d$-dimensional vectors with $d \geq 1$. For i.i.d. $\zeta_j$, $j = 1, \ldots, d$

$$\mathbb{E}\left[\exp\left(\zeta^T x_i\right)\right] = \mathbb{E}\left[\exp\left(\sum_{j=1}^d \zeta_j x_{ij}\right)\right]$$

$$= \mathbb{E}\left[\prod_{j=1}^d \exp\left(\zeta_j x_{ij}\right)\right]$$

$$= \prod_{j=1}^d \mathbb{E}\left[\exp\left(\zeta_j x_{ij}\right)\right],\quad\text{(independence)}$$

which is the moment generating function for random variable $\zeta_j$ evaluated at $t = x_{ij}$. Now for $\zeta_j \overset{\text{iid}}{\sim} \mathcal{L}(\mu, \rho)$

$$\prod_{j=1}^d \mathbb{E}\left[\exp\left(\zeta_j x_{ij}\right)\right] = \prod_{j=1}^d \frac{\exp\left(\mu x_{ij}\right)}{1 - \rho^2 x_{ij}^2},\text{ for }|x_{ij}| < 1/\rho\quad\forall j$$

$$= \frac{\exp\left(\mu\sum_{j=1}^d x_{ij}\right)}{\prod_{j=1}^d\left(1 - \rho^2 x_{ij}^2\right)},\text{ for }|x_{ij}| < 1/\rho\quad\forall j.$$

as a result

$$b(x_i) = \frac{\prod_{j=1}^d\left(1 - \rho^2 x_{ij}^2\right)}{\exp\left(\mu\sum_{j=1}^d x_{ij}\right)},\text{ with }|x_{ij}| < 1/\rho\quad\forall j \tag{12}$$

**The variance** Of interest to the performance of such an approach are the terms

$$\text{Var}_{\zeta \sim p_\zeta} \left[ b(x_i) \exp(\zeta^T x_i) \right] = b(x_i)^2 \text{Var}_{\zeta \sim p_\zeta} \left[ \exp(\zeta^T x_i) \right]$$

$$= b(x_i)^2 \left( \mathbb{E}_{\zeta \sim p_\zeta} \left[ \exp(\zeta^T x_i)^2 \right] - \mathbb{E}_{\zeta \sim p_\zeta} \left[ \exp(\zeta^T x_i) \right]^2 \right)$$

$$= b(x_i)^2 \left( \mathbb{E}_{\zeta \sim p_\zeta} \left[ \exp(2\zeta^T x_i) \right] - \mathbb{E}_{\zeta \sim gp_\zeta} \left[ \exp(\zeta^T x_i) \right]^2 \right)$$

$$= \frac{\prod_{j=1}^{d} \left( 1 - \rho^2 x_{ij}^2 \right)^2}{\exp \left( 2\mu \sum_{j=1}^{d} x_{ij} \right)} \left( \frac{\exp \left( 2\mu \sum_{j=1}^{d} x_{ij} \right)}{\prod_{j=1}^{d} \left( 1 - 4b^2 x_{ij}^2 \right)} - \frac{\exp \left( 2\mu \sum_{j=1}^{d} x_{ij} \right)}{\prod_{j=1}^{d} \left( 1 - \rho^2 x_{ij}^2 \right)^2} \right)$$

$$= \prod_{j=1}^{d} \frac{\left( 1 - \rho^2 x_{ij}^2 \right)^2}{\left( 1 - 4b^2 x_{ij}^2 \right)} - 1$$

with $|x_{ij}| < 1/2\rho \quad \forall j$, and

$$\left( \frac{b(x)b(x')}{b(x+x')} - 1 \right) = \frac{\frac{\prod_{j=1}^{d} \left( 1 - \rho^2 x_j^2 \right)}{\exp \left( \mu \sum_{j=1}^{d} x_j \right)} \frac{\prod_{j=1}^{d} \left( 1 - \rho^2 x_j'^2 \right)}{\exp \left( \mu \sum_{j=1}^{d} x_j' \right)}}{\frac{\prod_{j=1}^{d} \left( 1 - \rho^2 (x_j + x_j')^2 \right)}{\exp \left( \mu \sum_{j=1}^{d} (x_j + x_j') \right)}} - 1, \text{ with } |x_j|, |x_j'| \text{ and } |x_j + x_j'| < 1/\rho \quad \forall j$$

$$= \frac{\prod_{j=1}^{d} \left( 1 - \rho^2 x_j^2 \right) \left( 1 - \rho^2 x_j'^2 \right)}{\prod_{j=1}^{d} \left( 1 - \rho^2 (x_j + x_j')^2 \right)} - 1.$$

### B.2.2 Special Case 2: Gaussian Noise

Recall that $x_i$ and $\zeta$ are $d$-dimensional vectors with $d \geq 1$. The reciprocal of the bias correction

$$\frac{1}{b(x_i)} = \mathbb{E}_\zeta [\exp \left( \zeta^T x_i \right)],$$

is the moment generating function of random variable $\zeta^T x_i$ evaluated at $t = 1$. Now if $\zeta_j \stackrel{\text{iid}}{\sim} \mathcal{N}(\mu, \sigma^2)$, $j = 1, \ldots, d$, then

$$\zeta^T x_i = \sum_{j=1}^{d} \zeta_j x_{ij} \sim \mathcal{N}(\mu \sum_{j=1}^{d} x_{ij}, \sigma^2 \sum_{j=1}^{d} x_{ij}^2)$$

and therefore

$$\mathbb{E}_\zeta \left[ \exp \left( \zeta^T x_i \right) \right] = \exp \left( \mu \sum_{j=1}^{d} x_{ij} + \frac{1}{2} \sigma^2 \sum_{j=1}^{d} x_{ij}^2 \right).$$

**The variance** Of interest to the performance of such an approach are the terms

$$\text{Var}_{\zeta \sim p_\zeta} \left[ b(x_i) \exp(\zeta^T x_i) \right] = b(x_i)^2 \text{Var}_{\zeta \sim p_\zeta} \left[ \exp(\zeta^T x_i) \right]$$

$$= b(x_i)^2 \left( \mathbb{E}_{\zeta \sim p_\zeta} \left[ \exp(2\zeta^T x_i) \right] - \mathbb{E}_{\zeta \sim p_\zeta} \left[ \exp(\zeta^T x_i) \right]^2 \right)$$

$$= \exp \left( -2\mu \sum_{j=1}^{d} x_{ij} - \sigma^2 \sum_{j=1}^{d} x_{ij}^2 \right) \left( \exp \left( 2\mu \sum_{j=1}^{d} x_{ij} + 2\sigma^2 \sum_{j=1}^{d} x_{ij}^2 \right) \right.$$

$$\left. - \exp \left( 2\mu \sum_{j=1}^{d} x_{ij} + \sigma^2 \sum_{j=1}^{d} x_{ij}^2 \right) \right)$$

$$= \exp \left( \sigma^2 \sum_{j=1}^{d} x_{ij}^2 \right) - 1$$

and

$$\left(\frac{b(x)b(x')}{b(x+x')} - 1\right) = \frac{\exp\left(-\mu \sum_{j=1}^d x_j - \frac{1}{2}\sigma^2 \sum_{j=1}^d x_j^2\right) \exp\left(-\mu \sum_{j=1}^d x_j' - \frac{1}{2}\sigma^2 \sum_{j=1}^d x_j'^2\right)}{\exp\left(-\mu \sum_{j=1}^d (x_j + x_j') - \frac{1}{2}\sigma^2 \sum_{j=1}^d (x_j + x_j')^2\right)} - 1$$

$$= \exp\left(\frac{1}{2}\sigma^2 \sum_{j=1}^d \left\{(x_j + x_j')^2 - x_j^2 - x_j'^2\right\}\right) - 1$$

$$= \exp\left(\sigma^2 \sum_{j=1}^d x_j x_j'\right) - 1$$

### B.2.3 Differential Privacy

The differential privacy of the approach follows from the post-processing theorem: since the synthetic data $x_1, \ldots, x_{N_G}$ is already privatised, the corresponding weights $\bar{w}(x_1), ..., \bar{w}(x_{N_G})$ are $(\epsilon, \delta)$ differentially private, and the adversary can be assumed to know which differential privacy mechanism is used (Balle and Wang, 2018), the data curator can debias the weights without any additional privacy budget.

### B.2.4 Variance Comparison of Debiasing Ji & Elkan (2013)

Ji and Elkan (2013) provide bounds for the asymptotic variance of their privatised estimator. Here, we investigate the finite sample variance of their (biased) method and compare it with the finite variance of our unbiased estimator form Proposition 2. Note that we do not consider self-normalised IW while this is an implicit assumption made by Ji and Elkan (2013).

The variance of estimator $I_N(h|\overline{w})$, where $\overline{w}$ is defined in Equation (9), is given by

$$\mathrm{Var}_{p_G^*}\left[I_N(h|\overline{w})\right] = \frac{1}{N^2}\sum_{i=1}^N \mathrm{Var}_{p_G^*}\left[h(x_i)\overline{w}(x_i)\right] + \frac{2}{N^2}\sum_{i=1}^N \sum_{j<i} \mathrm{Cov}_{p_G^*}\left[h(x_i)\overline{w}(x_i), h(x_j)\overline{w}(x_j)\right]$$

$$= \frac{\overline{\sigma}^2(h)}{N} + \left(1 - \frac{1}{N}\right)\overline{c}(h).$$

where, $x, x' \sim p_G^*$. The term $\overline{\sigma}^2(h)$ is

$$\overline{\sigma}^2(h) = \mathrm{Var}_{p_G^*}\left[h(x)\widehat{w}(x)\exp(\zeta^T x)\right]$$

$$= \mathbb{E}_{p_G^*}\left[\left(h(x)\widehat{w}(x)\exp(\zeta^T x)\right)^2\right] - \mathbb{E}_{p_G^*}\left[h(x)\widehat{w}(x)\exp(\zeta^T x)\right]^2$$

$$= \mathbb{E}_{x\sim p_G}\left[h(x)^2\widehat{w}(x)^2 \mathbb{E}_{\zeta\sim p_\zeta}\left[\exp(\zeta^T x)^2\right]\right] - \mathbb{E}_{x\sim p_G(x)}\left[\frac{h(x)\widehat{w}(x)}{b(x)}\right]^2$$

$$= \mathbb{E}_{x\sim p_G}\left[h(x)^2\widehat{w}(x)^2\left(\mathrm{Var}_{\zeta\sim p_\zeta}\left[\exp(\zeta^T x)\right] + \frac{1}{b(x)^2}\right)\right] - \mathbb{E}_{x\sim p_G(x)}\left[\frac{h(x)\widehat{w}(x)}{b(x)}\right]^2$$

$$= \mathbb{E}_{x\sim p_G}\left[h(x)^2\widehat{w}(x)^2 \mathrm{Var}_{\zeta\sim p_\zeta}\left[\exp(\zeta^T x)\right]\right] + \mathrm{Var}_{x\sim p_G(x)}\left[\frac{h(x)\widehat{w}(x)}{b(x)}\right].$$

Further, $\overline{c}(h)$ is

$$\overline{c}(h) = \mathrm{Cov}_{p_G^*}\left[h(x)\widehat{w}(x)\exp\left(\zeta^T x\right), h(x')\widehat{w}(x')\exp\left(\zeta^T x'\right)\right]$$

$$= \mathbb{E}_{x,x'\sim p_G^*}\left[h(x)\widehat{w}(x)\exp\left(\zeta^T x\right) \cdot h(x')\widehat{w}(x')\exp\left(\zeta^T x'\right)\right]$$

$$- \mathbb{E}_{x\sim p_G^*}\left[h(x)\widehat{w}(x)\exp\left(\zeta^T x\right)\right] \cdot \mathbb{E}_{x'\sim p_G^*}\left[h(x')\widehat{w}(x')\exp\left(\zeta^T x'\right)\right],$$

where firstly,

$$\mathbb{E}_{x\sim p_G^*}\left[h(x)\widehat{w}(x)\exp\left(\zeta^T x\right)\right] \cdot \mathbb{E}_{x'\sim p_G^*}\left[h(x')\widehat{w}(x')\exp\left(\zeta^T x'\right)\right] = \mathbb{E}_{x\sim p_G(x)}\left[\frac{h(x)\widehat{w}(x)}{b(x)}\right]^2,$$

and

$$\mathbb{E}_{x,x'\sim p_G^*}\left[h(x)\widehat{w}(x)\exp\left(\zeta^T x\right)\cdot h(x')\widehat{w}(x')\exp\left(\zeta^T x'\right)\right]$$

$$=\mathbb{E}_{x,x'\sim p_G}\left[h(x)\widehat{w}(x)h(x')\widehat{w}(x')\mathbb{E}_{\zeta\sim p_\zeta}\left[\exp\left(\zeta^T(x+x')\right)\right]\right]$$

$$=\mathbb{E}_{x,x'\sim p_G}\left[h(x)\widehat{w}(x)h(x')\widehat{w}(x')\frac{1}{b(x+x')}\right]$$

$$=\mathbb{E}_{x,x'\sim p_G}\left[h(x)\widehat{w}(x)h(x')\widehat{w}(x')\left(\frac{1}{b(x+x')}-\frac{1}{b(x)b(x')}\right)\right]$$

$$+\mathbb{E}_{x,x'\sim p_G}\left[\frac{h(x)\widehat{w}(x)}{b(x)}\frac{h(x')\widehat{w}(x')}{b(x')}\right]$$

$$=\mathbb{E}_{x,x'\sim p_G}\left[h(x)\widehat{w}(x)h(x')\widehat{w}(x')\left(\frac{1}{b(x+x')}-\frac{1}{b(x)b(x')}\right)\right]$$

$$+\mathbb{E}_{x\sim p_G}\left[\frac{h(x)\widehat{w}(x)}{b(x)}\right]\mathbb{E}_{x'\sim p_G}\left[\frac{h(x')\widehat{w}(x')}{b(x')}\right]\quad\text{(indep.)}$$

$$=\mathbb{E}_{x,x'\sim p_G}\left[h(x)\widehat{w}(x)h(x')\widehat{w}(x')\left(\frac{1}{b(x+x')}-\frac{1}{b(x)b(x')}\right)\right]$$

$$+\mathbb{E}_{x\sim p_G}\left[\frac{h(x)\widehat{w}(x)}{b(x)}\right]^2$$

as a result

$$\overline{c}(h)=\mathbb{E}_{x,x'\sim p_G}\left[h(x)\widehat{w}(x)h(x')\widehat{w}(x')\left(\frac{1}{b(x+x')}-\frac{1}{b(x)b(x')}\right)\right]$$

$$=\mathbb{E}_{x,x'\sim p_G}\left[\frac{h(x)\widehat{w}(x)}{b(x)}\frac{h(x')\widehat{w}(x')}{b(x')}\left(\frac{b(x)b(x')}{b(x+x')}-1\right)\right].$$

**Comparisons after debiasing:** We can compare the variance of $I_N(h|\overline{w})$ with the previously evaluated variance of $I_N(h|\overline{w}^*)$ as follows

$$\mathrm{Var}_{p_G^*}\left[I_N(h|\overline{w}^*)\right]=\frac{\overline{\sigma}^{*2}(h)}{N}+\left(1-\frac{1}{N}\right)\overline{c}^*(h).$$

$$\mathrm{Var}_{P_G^*}\left[I_N(h|\overline{w})\right]=\frac{\overline{\sigma}^2(h)}{N}+\left(1-\frac{1}{N}\right)\overline{c}(h).$$

with

$$\overline{\sigma}^{*2}(h)=\mathbb{E}_{x\sim p_G}\left[h(x)^2\widehat{w}(x)^2\mathrm{Var}_{\zeta\sim p_\zeta}\left[b(x)\exp(\zeta x)\right]\right]+\mathrm{Var}_{x\sim p_G(x)}\left[h(x)\widehat{w}(x)\right]$$

$$\overline{\sigma}^2(h)=\mathbb{E}_{x\sim p_G}\left[h(x)^2\widehat{w}(x)^2\mathrm{Var}_{\zeta\sim p_\zeta}\left[\exp(\zeta^T x)\right]\right]+\mathrm{Var}_{x\sim p_G(x)}\left[\frac{h(x)\widehat{w}(x)}{b(x)}\right]$$

and

$$\overline{c}^*(h)=\mathbb{E}_{x,x'\sim p_G}\left[h(x)\widehat{w}(x)h(x')\widehat{w}(x')\left(\frac{b(x)b(x')}{b(x+x')}-1\right)\right]$$

$$\overline{c}(h)=\mathbb{E}_{x,x'\sim p_G}\left[\frac{h(x)\widehat{w}(x)}{b(x)}\frac{h(x')\widehat{w}(x')}{b(x')}\left(\frac{b(x)b(x')}{b(x+x')}-1\right)\right].$$

**Comparison for the introduction of Laplace noise:** From Equation (12), under $\zeta_j\sim\mathcal{L}(0,\rho)$ we have that

$$b(x_i)=\prod_{j=1}^{p}\left(1-\rho^2 x_{ij}^2\right),\text{ with }|x_{ij}|<1/\rho\quad\forall j.$$

The condition that $|x_{ij}| < 1/\rho$ ensures that

$$0 \leq \left(1 - \rho^2 x_{ij}^2\right) \leq 1, \quad \forall j$$

$$\Rightarrow 0 \leq b(x) = \prod_{j=1}^{p} \left(1 - \rho^2 x_j^2\right) \leq 1$$

As a result,

$$\text{Var}_{\zeta \sim g}\left[b(x)\exp(\zeta^T x)\right] \leq \text{Var}_{\zeta \sim g}\left[\exp(\zeta^T x)\right], \quad \forall x$$

$$\text{and } h(x)\widehat{w}(x) \leq \frac{h(x)\widehat{w}(x)}{b(x)}, \quad \forall x$$

which provides that

$$\overline{\sigma}^{*2}(h) \leq \overline{\sigma}^2(h)$$

$$\text{and } \overline{c}^*(h) \leq \overline{c}(h)$$

$$\Rightarrow \text{Var}_{p_G^*}\left[I_N(h|\overline{w}^*)\right] \leq \text{Var}_{p_G^*}\left[I_N(h|\overline{w})\right]. \tag{13}$$

Not only does debiasing remove bias, it also makes the estimator's variance smaller.

## B.3   THEOREM 2: NOISY IMPORTANCE SAMPLING

For privacy purposes, we want to be able to noise the importance weights as in

$$\log w^*(x) = \log \widehat{w}(x) + \zeta, \text{ for } \zeta \sim g \text{ drawn from a noise distribution} \tag{14}$$

but we would like to still preserve the consistency properties of importance sampling estimates.

To achieve this, we expand the original target in importance sampling as follows

$$p_D^*(x, \zeta) = p_D(x)\exp(\zeta)g(\zeta)$$

where $\zeta \in \mathbb{R}$ will correspond to some additive noise on the log weights, and $g(\zeta)$ is a probability density on $\mathbb{R}$ such that by assumption

$$\int \exp(\zeta)g(\zeta)d\zeta = 1,$$

So, in particular, this implies that

$$\int p_D^*(x, \zeta)d\zeta = p_D(x).$$

Now, we can use a proposal density $p_G^*(x, \zeta) = p_G(x)g(\zeta)$ targeting $p_D^*(x, \zeta)$ and the resulting importance weight is indeed

$$w^*(x, \zeta) = \frac{p_D^*(x, \zeta)}{p_G^*(x, \zeta)} = \widehat{w}(x)\exp(\zeta),$$

i.e. the importance weight in this extended space is a noisy version of the original weight $\widehat{w}(x)$. We thus have

$$\mathbb{E}_{p_D}[h(x)] = \mathbb{E}_{p_G}[h(x)\widehat{w}(x)]$$

$$= \mathbb{E}_{p_G^*}[h(x)w^*(x, \zeta)]$$

$$= \mathbb{E}_{p_G^*}[h(x)\widehat{w}(x)\exp(\zeta)].$$

It follows that for i.i.d. $(x_i, \zeta_i) \sim p_G^*$, i.e. $x_i \sim p_G$ and $\zeta_i \sim g$, then

$$I_N(h|w^*) = \frac{1}{N}\sum_{i=1}^{N} h(x_i)\widehat{w}(x_i)\exp(\zeta_i)$$

is an unbiased and consistent estimator of $\mathbb{E}_{p_D}[h(x)]$. Its variance is

$$\text{Var}\left[I_N(h|w^*)\right] = \frac{1}{N}\text{Var}_{p_D^*}\left[h(x)\widehat{w}(x)\exp(\zeta)\right] = \frac{\sigma^{*2}(h)}{N}.$$

By the variance decomposition formula, we have

$$
\begin{aligned}
\sigma^{*2}(h) =& \text{Var}_{p_D^*}\left[h(x)\widehat{w}(x)\exp(\zeta)\right]\\
=& \mathbb{E}_g\left[\exp(\zeta)\right]^2 \text{Var}_{p_G}\left[h(x)\widehat{w}(x)\right]\\
& + \text{Var}_g\left[\exp(\zeta)\right]\mathbb{E}_{p_G}\left[(h(x)\widehat{w}(x))^2\right] \quad \text{(variance decomposition formula)}\\
=& \sigma^2(h) + \text{Var}_g\left[\exp(\zeta)\right]\mathbb{E}_{p_G}[(h(x)\widehat{w}(x))^2],
\end{aligned}
$$

as $\mathbb{E}_g\left[\exp(\zeta)\right] = 1$ by assumption and $\text{Var}\left[I_N(h|w)\right] = \frac{1}{N}\text{Var}_{p_G}\left[h(x)\widehat{w}(x)\right]$. The variance of our estimator is inflated as expected by the introduction of noise.

## B.4 COROLLARY 1 AND 2: DIFFERENTIAL PRIVACY OF LOG-LAPLACE NOISED IMPORTANCE WEIGHTS

Following Kozubowski and Podgórski (2003), the (symmetric) log-Laplace distribution is the distribution of random variable $x$ such that $y = \log(x)$ has a Laplace density with location parameter $\mu$ and scale $\lambda$. The density of a log-Laplace$(\mu, \lambda)$ random variable is

$$f_X(x|\mu, \lambda) = \frac{1}{2\lambda}\frac{1}{x}\exp\left(-\frac{1}{\lambda}\left|\log x - \mu\right|\right).$$

Note this is recovered from the asymmetric log-Laplace in Kozubowski and Podgórski (2003) with $\alpha = \beta = \frac{1}{\lambda}$. Kozubowski and Podgórski (2003) further provide forms for the expectation and variance of the log-Laplace distribution as

$$\mathbb{E}\left[X\right] = \frac{\exp(\mu)}{1 - \lambda^2} \text{ for } \lambda < 1, \tag{15}$$

$$\text{Var}[X] = \exp(2\mu)\left(\frac{1}{1 - 4\lambda^2} - \frac{1}{(1-\lambda^2)^2}\right) \text{ for } \lambda < \frac{1}{2}.$$

Next we wish to investigate the differential privacy provided by using the Laplace mechanism (Dwork et al., 2006) to noise importance weights. Adding Laplace noise to the log-weights, as in Equation (14), is equivalent to multiplying the importance weights by log-Laplace noise. In order for the importance sampling to remain unbiased, the log-Laplace noise must have expectation 1. From Equation (15) this will be the case for all $\lambda < 1$ if we set $\mu = \log\left(1 - \lambda^2\right)$.

A binary logistic-regression classifier specifies class probabilities

$$\widehat{p}(y = 1|x, \widehat{\beta}) = \frac{1}{1 + \exp\left(-x\widehat{\beta}\right)}, \quad \widehat{p}(y = 0|x, \widehat{\beta}) = \frac{\exp\left(-x\widehat{\beta}\right)}{1 + \exp\left(-x\widehat{\beta}\right)}.$$

We denote by $z_{1:N_G}$ the private data sampled from the DGP, and by $x_{1:N_D}$ the synthetic data sampled from the SDGP. Let $z'_{1:N_G}$ be the neighboring data set of $z_{1:N_G}$. The importance weights estimated by such a classifier become

$$
\begin{aligned}
\widehat{w}(x_i|x_{1:N_G}, z_{1:N_D}) =& \frac{\tilde{p}(y_i = 1|x_i, \hat{\beta}(x_{1:N_G}, z_{1:N_D}))}{\tilde{p}(y_i = 0|x_i, \hat{\beta}(x_{1:N_G}, z_{1:N_D}))}\frac{N_D}{N_G}\\
=& \frac{1}{1 + \exp\left(-x_i\hat{\beta}(x_{1:N_G}, z_{1:N_D})\right)}\frac{1 + \exp\left(-x_i\hat{\beta}(x_{1:N_G}, z_{1:N_D})\right)}{\exp\left(-x_i\hat{\beta}(x_{1:N_G}, z_{1:N_D})\right)}\frac{N_D}{N_G}\\
=& \exp\left(x_i\hat{\beta}(x_{1:N_G}, z_{1:N_D})\right)\frac{N_D}{N_G},
\end{aligned}
$$

and as a result

$$\left| \log \widehat{w}(x_i | x_{1:N_G}, z_{1:N_D}) - \log \widehat{w}(x_i | x_{1:N_G}, z'_{1:N_D}) \right|$$

$$= \left| x_i \hat{\beta}(x_{1:N_G}, z_{1:N_D}) + \log \frac{N_D}{N_G} - \left( x_i \hat{\beta}(x_{1:N_G}, z'_{1:N_D}) + \log \frac{N_D}{N_G} \right) \right|$$

$$= \left| x_i \hat{\beta}(x_{1:N_G}, z_{1:N_D}) - x_i \hat{\beta}(x_{1:N_G}, z'_{1:N_D}) \right|$$

$$= \left| \sum_{j=1}^{p} x_{ij} \left( \hat{\beta}(x_{1:N_G}, z_{1:N_D})_j - \hat{\beta}(x_{1:N_G}, z'_{1:N_D})_j \right) \right|$$

$$\leq |x_i| \sum_{j=1}^{d} \left| \left( \hat{\beta}(x_{1:N_G}, z_{1:N_D})_j - \hat{\beta}(x_{1:N_G}, z'_{1:N_D})_j \right) \right|$$

$$\leq \frac{2\sqrt{d}}{N_D \lambda}$$

if the features are minmax scaled using the sensitivity computed by Chaudhuri et al. (2011).

## B.5 REMARK 1: THE IMPORTANCE-WEIGHTED LIKELIHOOD AND *M*-ESTIMATION

**Remark 1.** *Minimisation of the importance weight adjusted log-likelihood, $-w(x_i) \log f(x_i|\theta)$, can be viewed as an $M$-estimator with clear relations to the standard MLE.*

Remark 1 of the paper points out the the connection between the Minimisation of the importance weight adjusted log-likelihood, $\ell_{IW}(x, \theta) := -w(x_i) \log f(x_i|\theta)$ and the standard maximum likelihood estimator which can be seen through the lens of M-estimation. We exemplify this below.

Following Van der Vaart (2000), the $M$-estimate of parameter

$$\beta_h^* := \arg\max_{\beta} \mathbb{E}_{x \sim p_D} \left[ h(\beta, x) \right]$$

is given by

$$\hat{\beta}_h^{(n)} := \arg\max_{\beta} \sum_{i=1}^{n} h(\beta, x_i).$$

The estimator $\hat{\beta}_h^{(n)}$ is consistent and is asymptotically normal, i.e.

$$\sqrt{n} \left( \hat{\beta}_h^{(n)} - \beta_h^* \right) \xrightarrow{D} \mathcal{N} \left( 0, \tilde{V}(\beta_h^*) \right)$$

where

$$\tilde{V}(\beta) := \left( \mathbb{E} \left[ \nabla_{\beta}^2 h(\beta, x) \right] \right)^{-1} \cdot \text{Var} \left[ \nabla_{\beta} h(\beta, x) \right] \cdot \left( \mathbb{E} \left[ \nabla_{\beta}^2 h(\beta, x) \right] \right)^{-1}.$$

M-estimators generalises the case of MLE under model misspecification and the variance calculation collapses to the standard inverse Fisher's information if the likelihood is correctly specified for the DGP.

The minimiser of the importance weight adjusted log-likelihood can be considered an M-estimate with the following form

$$\hat{\theta}_{IW}^{(n)} = \arg\max \{-\ell_{IW}(x; \theta)\} = \arg\max \{w(x) \log f(x; \theta)\}.$$

As a result, given $x_{1:n} \sim P_G$ the covariance of the asymptotic Gaussian distribution for $\hat{\theta}_{IW}^{(n)}$ simplifies to,

$$\tilde{V}_{IW}(\theta_{IW}^*) = \left( \mathbb{E}_{p_G} \left[ -\nabla_{\theta}^2 \ell_{IW}(x, \theta_{IW}^*) \right] \right)^{-1} \cdot \text{Var}_{p_G} \left[ -\nabla_{\theta} \ell_{IW}(x, \theta_{IW}^*) \right] \cdot \left( \mathbb{E}_{p_G} \left[ -\nabla_{\theta}^2 \ell_{IW}(x, \theta_{IW}^*) \right] \right)^{-1}$$

$$= \left( \mathbb{E}_{p_D} \left[ -\nabla_{\theta}^2 \ell_0(x, \theta_0^*) \right] \right)^{-1} \cdot \text{Var}_{p_G} \left[ -\nabla_{\theta} \ell_{IW}(x, \theta_{IW}^*) \right] \cdot \left( \mathbb{E}_{p_D} \left[ -\nabla_{\theta}^2 \ell_0(x, \theta_0^*) \right] \right)^{-1}$$

$$= \left( \mathbb{E}_{p_D} \left[ -\nabla_{\theta}^2 \ell_0(x, \theta_0^*) \right] \right)^{-1} \cdot \mathbb{E}_{p_G} \left[ \left( -\nabla_{\theta} \ell_{IW}(x, \theta_{IW}^*) \right) \left( -\nabla_{\theta} \ell_{IW}(x, \theta_{IW}^*) \right)^T \right] \cdot \left( \mathbb{E}_{p_D} \left[ -\nabla_{\theta}^2 \ell_0(x, \theta_0^*) \right] \right)^{-1}$$

where $\text{Var}_{p_G} \left[ -\nabla_{\theta} \ell_{IW}(x, \theta_{IW}^*) \right] = \mathbb{E}_{p_G} \left[ \left( -\nabla_{\theta} \ell_{IW}(x, \theta_{IW}^*) \right) \left( -\nabla_{\theta} \ell_{IW}(x, \theta_{IW}^*) \right)^T \right]$ because at the maximiser $\theta_{IW}^*$ $\mathbb{E}_{p_G} \left[ -\nabla_{\theta} \ell_{IW}(x, \theta_{IW}^*) \right] = 0$

Further we can write the variance of the minimiser of the importance weight adjusted log-likelihood in terms of the variance of the standard MLE given the same number of observations $x_{1:n} \sim P_D$ as follows:

$$\frac{\tilde{V}_{IW}\left(\theta_{IW}^*\right)}{\tilde{V}_0\left(\theta_0^*\right)} = \frac{\mathbb{E}_{P_G}\left[\left(\nabla_\theta \ell_{IW}(x, \theta_{IW}^*)\right)\left(\nabla_\theta \ell_{IW}(x, \theta_{IW}^*)\right)^T\right]}{\mathbb{E}_{P_D}\left[\left(\nabla_\theta \ell_0(x, \theta_0^*)\right)\left(\nabla_\theta \ell_0(x, \theta_0^*)\right)^T\right]} = \frac{\mathbb{E}_{P_D}\left[w(x)\left(\nabla_\theta \ell_0(x, \theta_{IW}^*)\right)\left(\nabla_\theta \ell_0(x, \theta_{IW}^*)\right)^T\right]}{\mathbb{E}_{P_D}\left[\left(\nabla_\theta \ell_0(x, \theta_0^*)\right)\left(\nabla_\theta \ell_0(x, \theta_0^*)\right)^T\right]}.$$

We can then use such notions to produce an idea of the effective sample size of synthetic data.

### B.5.1 The Effective Sample Size of Synthetic Data

When constructing traditional Importance Sampling estimates it is typical to talk about the 'effective sample' size of the sample from the proposal density. The effective sample size is the number of independent samples from the true target that gives an unbiased estimator with the same variance as the importance sampling estimator using $N_G$ samples from the proposal density. When using importance weights to adjust the likelihood for Bayesian updating we are not directly seeking to estimate an expectation, but minimize an (expected) loss to produce a parameter estimate.

Analogously, in this scenario we define the effective sample size of the synthetic data as the number of samples, $N_G^{(e)}$, from true DGP $P_D$ that would provide an unbiased maximum likelihood estimate (MLE) with the same variance as the Importance-Weighted MLE (IW-MLE), i.e.

$$N_G^{(e)} := \left\{n : \left|V\left[\hat{\theta}_{IW}^{(N_G)}\right]\right| = \left|V\left[\hat{\theta}_0^{(n)}\right]\right|\right\},$$

where the function $V$ corresponds to the asymptotic variance of that estimator, and $|\cdot|$ is a norm summary of the matrix values covariance of the estimator. Given the asymptotic analysis presented above for the importance-weighted likelihood we have that

$$N_G^{(e)} = \left(\frac{\sqrt{N_G}\left|\tilde{V}\left(\hat{\theta}_0^{(n)}\right)\right|}{\left|\tilde{V}\left(\hat{\theta}_{IW}^{(N_G)}\right)\right|}\right)^2 \tag{16}$$

where

$$\frac{\left|\tilde{V}\left(\hat{\theta}_0^{(n)}\right)\right|}{\left|\tilde{V}\left(\hat{\theta}_{IW}^{(N_G)}\right)\right|} = \frac{\left|\mathbb{E}_{P_D}\left[\widehat{w}(x)\left(\nabla_\theta \ell_0(x, \theta_{IW}^*)\right)\left(\nabla_\theta \ell_0(x, \theta_{IW}^*)\right)^T\right]\right|}{\left|\mathbb{E}_{P_D}\left[\left(\nabla_\theta \ell_0(x, \theta_0^*)\right)\left(\nabla_\theta \ell_0(x, \theta_0^*)\right)^T\right]\right|}$$

$$= \frac{\left|\mathbb{E}_{P_G}\left[\left(\nabla_\theta \ell_{IW}(x, \theta_{IW}^*)\right)\left(\nabla_\theta \ell_{IW}(x, \theta_{IW}^*)\right)^T\right]\right|}{\left|\mathbb{E}_{P_G}\left[\widehat{w}(x)\left(\nabla_\theta \ell_0(x, \theta_0^*)\right)\left(\nabla_\theta \ell_0(x, \theta_0^*)\right)^T\right]\right|}.$$

We note that for multidimensional parameter vectors the $V$'s are covariance matrices and therefore we need to take a scalar summary using the norm $|\cdot|$ of these matrices in order to provide an integer effective sample size $N_G^{(e)}$. Faced with a similar problem Lyddon et al. (2018) consider the matrix trace for example.

Lastly, given a sample $x_{1:N_G} \sim P_G$ the effective sample size can be estimated by using empirical expectations

$$\frac{\left|\tilde{V}\left(\hat{\theta}_0^{(n)}\right)\right|}{\left|\tilde{V}\left(\hat{\theta}_{IW}^{(N_G)}\right)\right|} \approx \frac{\left|\frac{1}{N_G}\sum_{i=1}^{N_G}\left(\nabla_\theta \ell_{IW}(x_i, \hat{\theta}_{IW}^{(n)})\right)\left(\nabla_\theta \ell_{IW}(x_i, \hat{\theta}_{IW}^{(n)})\right)^T\right|}{\left|\frac{1}{N_G}\sum_{i=1}^{N_G}\widehat{w}(x_i)\left(\nabla_\theta \ell_0(x_i, \hat{\theta}_{IW}^{(n)})\right)\left(\nabla_\theta \ell_0(x_i, \hat{\theta}_{IW}^{(n)})\right)^T\right|}.$$

### B.6 THEOREM 1: ASYMPTOTIC POSTERIOR DISTRIBUTION OF IMPORTANCE WEIGHTED BAYESIAN UPDATING

Section 3.1 of the paper considers the importance weighted Bayesian updating as a special case of general Bayesian updating where the loss function is specifically chosen to account for the fact that inference is being done with samples from $p_G$ while trying to approximate $p_D$. We henceforth write

$$\pi_{IW}(\theta|\{x_i\}_{i\in\{1,\dots,N_G\}}) \propto \pi(\theta)\exp\left(-\sum_{i=1}^{N_G}-\widehat{w}(x_i)\log f(x_i|\theta)\right)$$

$$= \pi(\theta)\exp\left(-\sum_{i=1}^{N_G}\ell_{IW}(x_i;\theta)\right),$$

for $\ell_{IW}(x_i; \theta) := -\widehat{w}(x_i) \log f(x_i|\theta)$ and $\widehat{w}(x_i) = p_D(x_i)/p_G(x_i)$. The next theorem shows that such a posterior given observations from $p_G$ has the same asymptotic distribution as the standard Bayes posterior given samples from $p_D$ would have, and therefore we consider this posterior to be asymptotically calibrated.

We give here the formal statement of Theorem 1. Below $\xrightarrow{D}$ denotes convergence in distribution.

**Theorem 1.** *Let the regular conditions in (Chernozhukov and Hong, 2003; Lyddon et al., 2018) hold. Consider $\hat{\theta}_{IW}^{(N)} := \arg\min_{\theta \in \Theta} \sum_{i=1}^{N} \ell_{IW}(x_i; \theta)$, $x_i \overset{i.i.d.}{\sim} p_G$ and $\hat{\theta}_0^{(N)} := \arg\min_{\theta \in \Theta} \sum_{i=1}^{N} \ell_0(x_i; \theta)$, $x_i \overset{i.i.d.}{\sim} p_D$ where $\ell_0(x; \theta) := -\log f(x; \theta)$. Then both $\hat{\theta}_0^{(N)}$ and $\hat{\theta}_{IW}^{(N)}$ are consistent estimates of $\theta_0^* := \arg\min_{\theta \in \Theta} \int \ell_0(x; \theta) dP_D(x)$. Moreover there exists a non-singular matrix $J^{-1}$ such that we have under the importance weighted Bayesian posterior $\pi_{IW}(\theta|x_{1:N})$*

$$\sqrt{N}\left(\theta - \hat{\theta}_{IW}^{(N)}\right) \xrightarrow{D} \mathcal{N}\left(0, J^{-1}\right),$$

*almost surely w.r.t. $x_{1:\infty}$[1] while under the standard Bayesian posterior $\pi(\theta|x_{1:N})$*

$$\sqrt{N}\left(\theta - \hat{\theta}_0^{(N)}\right) \xrightarrow{D} \mathcal{N}\left(0, J^{-1}\right),$$

*almost surely w.r.t. $x_{1:\infty}$.*

**Proof.** Firstly, define

$$\theta_{IW}^* := \arg\min_{\theta \in \Theta} \int \ell_{IW}(x; \theta) dP_G(x), \quad J_{IW}(\theta) := \int \nabla_\theta^2 \ell_{IW}(x; \theta) dP_G(x).$$

Then Chernozhukov and Hong (2003); Lyddon et al. (2018) show that under regularity conditions the following asymptotic result holds

$$\sqrt{N}\left(\theta - \hat{\theta}_{IW}^{(N)}\right) \xrightarrow{D} \mathcal{N}\left(0, J_{IW}\left(\theta_{IW}^*\right)^{-1}\right)$$

as $N \to \infty$ when $\theta$ is distributed according to the general Bayesian posterior almost surely w.r.t. $x_{1:\infty}$. Similarly, if we define

$$J_0(\theta) := \int \nabla_\theta^2 \ell_0(x; \theta) dP_D(x),$$

then we have that under the standard Bayesian posterior (Chernozhukov and Hong, 2003; Kleijn et al., 2012; Lyddon et al., 2018)

$$\sqrt{N}\left(\theta - \hat{\theta}_0^{(N)}\right) \xrightarrow{D} \mathcal{N}\left(0, J_0\left(\theta_0^*\right)^{-1}\right)$$

almost surely w.r.t. $x_{1:\infty}$. Now it follows from the importance sampling identity that

$$\theta_{IW}^* = \arg\min_{\theta \in \Theta} \int \ell_{IW}(x; \theta) dP_G(x) = \arg\min_{\theta \in \Theta} \int \ell_0(x; \theta) dP_D(x) = \theta_0^*,$$

$$J_{IW}(\theta) = \int \nabla_\theta^2 \ell_{IW}(x; \theta) dP_G(x) = \int \widehat{w}(x) \nabla_\theta^2 \ell_0(x; \theta) dP_G(x) = \int \nabla_\theta^2 \ell_0(x; \theta) dP_D(x) = J_0(\theta)$$

Moreover $\hat{\theta}_0^{(N)}$ and $\hat{\theta}_{IW}^{(N)}$ are also consistent estimates of $\theta_0^*$ under the same regularity conditions. This establishes the result.

### B.6.1   Finite Sample Importance-Weighted Bayesian posterior

To complement the asymptotic results connecting the importance weighted general Bayesian posterior given data from $p_G$ and the standard Bayesian $p_D$ we can consider the difference between these two for finite $n = m$. This is formulated in the following proposition.

**Proposition 4.** *The expected KLD beween standard Bayesian posterior $\pi(\theta|x_{1:n})$ and its importance weighted approximation $\pi_{IW}(\theta|z_{1:m})$ in expectation over the generating distributions for $x_{1:n} \sim P_D$ and $z_{1:m} \sim P_G$, for $n = m$ is*

$$\mathbb{E}_{x \sim p_D}\left[\mathbb{E}_{z \sim p_G}\left[KLD(\pi(\theta|x_{1:n})||\pi_{IW}(\theta|z_{1:m}))\right]\right]$$
$$= n\mathbb{E}_{x \sim p_D}\left[\mathbb{E}_{\theta \sim \pi(\cdot|x_{1:n})}\left[\left(\log f(x; \theta) - \mathbb{E}_{x' \sim p_D}\left[\log f(x'; \theta)\right]\right)\right]\right]$$

---

[1] $\pi_{IW}(\theta|x_{1:N})$ and $\pi(\theta|x_{1:N})$ are here interpreted as random probability measures, and functions of the random observations $x_{1:N}$.

**Proof.** We have

$$\mathbb{E}_{x\sim p_D}\left[\mathbb{E}_{z\sim p_G}\left[KLD(\pi(\theta|x_{1:n})||\pi_{IW}(\theta|z_{1:m}))\right]\right]$$

$$=\mathbb{E}_{x\sim p_D}\left[\mathbb{E}_{z\sim p_G}\left[\int \pi(\theta|x_{1:n})\log\frac{\pi(\theta|x_{1:n})}{\pi_{IW}(\theta|z_{1:m})}d\theta\right]\right]$$

$$=\mathbb{E}_{x\sim p_D}\left[\mathbb{E}_{z\sim p_G}\left[\mathbb{E}_{\pi(\theta|x_{1:n})}\left[\sum_{i=1}^{n}\log f(x_i;\theta)-\sum_{j=1}^{m}\widehat{w}(z_i)\log f(z_i;\theta)\right]\right]\right].$$

Now by Fubini we can reorder these integrals assuming that they all exist

$$=\mathbb{E}_{x\sim p_D}\left[\mathbb{E}_{\theta\sim\pi(\cdot|x_{1:n})}\left[\left(\sum_{i=1}^{n}\log f(x_i;\theta)-\sum_{j=1}^{m}\mathbb{E}_{z\sim p_G}\left[\widehat{w}(z_i)\log f(z_i;\theta)\right]\right)\right]\right]$$

$$=\mathbb{E}_{x\sim p_D}\left[\mathbb{E}_{\theta\sim\pi(\cdot|x_{1:n})}\left[\left(\sum_{i=1}^{n}\log f(x_i;\theta)-m\mathbb{E}_{x'\sim p_D}\left[\log f(x';\theta)\right]\right)\right]\right].$$

Now assuming $n=m$, we have

$$=\mathbb{E}_{x\sim p_D}\left[\mathbb{E}_{\theta\sim\pi(\cdot|x_{1:n})}\left[\sum_{i=1}^{n}\left(\log f(x_i;\theta)-\mathbb{E}_{x'\sim p_D}\left[\log f(x';\theta)\right]\right)\right]\right]$$

$$=n\mathbb{E}_{x\sim p_D}\left[\mathbb{E}_{\theta\sim\pi(\cdot|x_{1:n})}\left[\left(\log f(x;\theta)-\mathbb{E}_{x'\sim p_D}\left[\log f(x';\theta)\right]\right)\right]\right].$$

# C  EXPERIMENTS

## C.1  EXPERIMENTAL DETAILS

Please refer to Table 4 for an overview of the data sets used. We considered a random 80/20 train test split for all data sets except for MNIST for which the default split was used.

| Data | # training observations | # features | prediction problem |
|---|---|---|---|
| Iris | 150 | 4 | 3-class classification |
| tgfb | 262 | 7 | regression |
| Boston | 506 | 10 | regression |
| Breast | 569 | 30 | binary classification |
| Banknote | 1372 | 4 | binary classification |
| MNIST | 60000 | 784 | 10-class classification |

Table 4: Characteristics of the analysed data sets

We obtained the code for PrivBayes from `https://github.com/DataResponsibly/DataSynthesizer`, and the code for DPCGAN from `https://github.com/ricardocarvalhods/dpcgan`. This code was used and changed to write the code for DPGAN. For the logistic regression alternatives we use an adaption of the `sklearn` implementation. DPGAN was trained on labelled data by concatenating the features with the one hot encoding of the labels. Our implementation will be made available online. We train different downstream tasks on the synthetic data and test them on test data to ensure their utility for the setting of supervised learning. The downstream algorithms were trained using `sklearn` with default parameters.

Hyperparameter tuning is a non-private operation as it queries private data to evaluate the model at validation time. To ensure that we do not undermine the performance of the baselines we tuned them for $\epsilon=1.$, and chose default parameters for our method. PrivBayes is trained in correlated attribute mode, and with optimal bandwidth computation. For the GAN alternatives, we tuned the norm clip (1.0, 0.5), the batch size (32, 64), and number of epochs (50, 100) with grid search on a validation set (10% split of training). The noise multiplier was chosen such that the desired privacy budget was reached. The models were then retrained on the full training data set. Note that these hyperparameters are chosen smaller than in a non-private setting as the noise to be added would otherwise explode. The optimal hyperparameters can be found in the GitHub repository. Further we chose learning rate of the discriminator and generator as 0.15, and the

number of hidden dimensions as $d$ following Jordon et al. (2019). For the MNIST experiment, we chose to use the hyperparameters found by Torkzadehmahani et al. (2019). The regularisation parameter of the logistic regression for weight estimation was chosen from $0.1, 1, 2$.

The MLP for likelihood ratio estimation was computed based on the `tensorflow` and `tensorflow_privacy` package. To ensure the privacy of the MLP, we started with a configuration of one epoch, a batch size of 1, an L2 norm clip of 1, a noise multiplier of 5.2, 20 microbatches and a learning rate of 0.1. We computed the $\epsilon$ using built-in functions and increased/decreased the noise multiplier and the number of epochs until the desired privacy level was reached. We chose $N_S = N_D$ unless otherwise mentioned. To compute the output-noised weights we computed the largest $N_S$ such that the scale restriction was satisfied and conducted the downstream analysis on this smaller dataset.

## C.2 COMPUTATIONAL TIME OF IMPORTANCE WEIGHT ESTIMATION

Please refer to Table 5 for an overview of the additional time needed to compute the importance weights. All experimental results were computed by training on a single Tesla V100 GPU. We observe that the estimation of the importance weights comes with negligible computational overhead.

| weighting | Iris | Banknote | Housing | Breast | MNIST |
|---|---|---|---|---|---|
| BetaNoised | $0.0064_{\pm 0.0002}$ | $0.0084_{\pm 0.0002}$ | $0.0133_{\pm 0.0011}$ | $0.0824_{\pm 0.0206}$ | $51.5605_{\pm 9.0042}$ |
| BetaDebiased | $0.0237_{\pm 0.0125}$ | $0.0112_{\pm 0.0003}$ | $0.0742_{\pm 0.0083}$ | $0.1856_{\pm 0.0858}$ | $59.0723_{\pm 10.5120}$ |
| DP-MLP | $0.8338_{\pm 0.0964}$ | $5.4649_{\pm 0.0654}$ | $1.7303_{\pm 0.1104}$ | $2.9363_{\pm 0.1208}$ | $87.2693_{\pm 4.7303}$ |
| Discriminator | $0.0000_{\pm 0.0000}$ | $0.0000_{\pm 0.0000}$ | $0.0000_{\pm 0.0000}$ | $0.0000_{\pm 0.0000}$ | $0.0000_{\pm 0.0001}$ |
| LogReg | $0.0071_{\pm 0.0004}$ | $0.0099_{\pm 0.0003}$ | $0.0143_{\pm 0.0012}$ | $0.0910_{\pm 0.0210}$ | $52.0331_{\pm 9.1285}$ |
| MLP | $0.7741_{\pm 0.1436}$ | $1.5895_{\pm 0.0261}$ | $1.7491_{\pm 0.1414}$ | $1.4480_{\pm 0.1441}$ | $30.1968_{\pm 6.3155}$ |

Table 5: Additional computational time in seconds needed for the computation of importance weights averaged over 10 seeds and SDGP for $\epsilon = 1$.

## C.3 CHOICE OF PRIVACY SPLIT

In Figure 3, we plot the change in evaluation metrics for different values of privacy budget splits. We notice that the impact of the split parameter decreases the larger $\epsilon$ is. Similarly, the variability in the metrics for different $\delta$ splits decreases, the larger $\epsilon_{IW}$ is, where $\epsilon_{IW}$ denotes the privacy budget dedicated to the importance weight estimation. While a larger $\delta$ split of 30-50% seems beneficial for DP-MLP, the fraction of $\epsilon$ dedicated to the importance weighting model should be chosen relatively small, i.e. 10%. Note that we chose these default values based on their performance on the Adult, Credit and Spam data set. Tuning them to the underlying data and task characteristics will be able to improve their results. As hyperparameter tuning is an unsolved problem in DP, we leave the procedure for choosing the optimal privacy split per data set for future work. We note that an additional intricacy appears in DP because of the noise injection which increases the variability of the model's performances.

## C.4 MSE OF IMPORTANCE WEIGHT ESTIMATION

For each of our experiments, we compute the mean squared error between the privatised parameters of the logistic regression for importance weight estimation and the parameters of an unperturbed logistic regression trained on the private data. Please refer to Table 6 for the results. We observe that debiasing almost always decreases the MSE in the low-privacy regimes. For large privacy budgets, the scale of the perturbations can be negligible for low-dimensional data sets which is why both approaches perform similarly on Iris and Banknote, but debiasing still helps with larger data sets such as Breast.

## C.5 BAYESIAN UPDATING EXPERIMENTAL DETAILS

In addition to the logistic regression ROC-AUC score distributions presented in the main body of the paper, we applied importance weighted posteriors to updating and learning the parameters of linear regression and multinomial logistic regression models applied to the TGFB and Iris datasets respectively, see Figures 4a and 4b. It can be seen that in the case of linear regression, the DP-MLP and MLP IW methods are again very effective, with the performance improving across all SDGPs. Other methods again tend to reduce variance in the results whilst not damaging performance and so can be seen to be effective in at least ensuring greater robustness and consistency when learning under synthetic data. In the case of the Iris data, we calculated 1 vs all ROC-AUC scores for each class separately, then averaged these per-class ROC-AUCs to get a single multi-class average ROC-AUC. Again, MLP and DP-MLP are stand-out in their

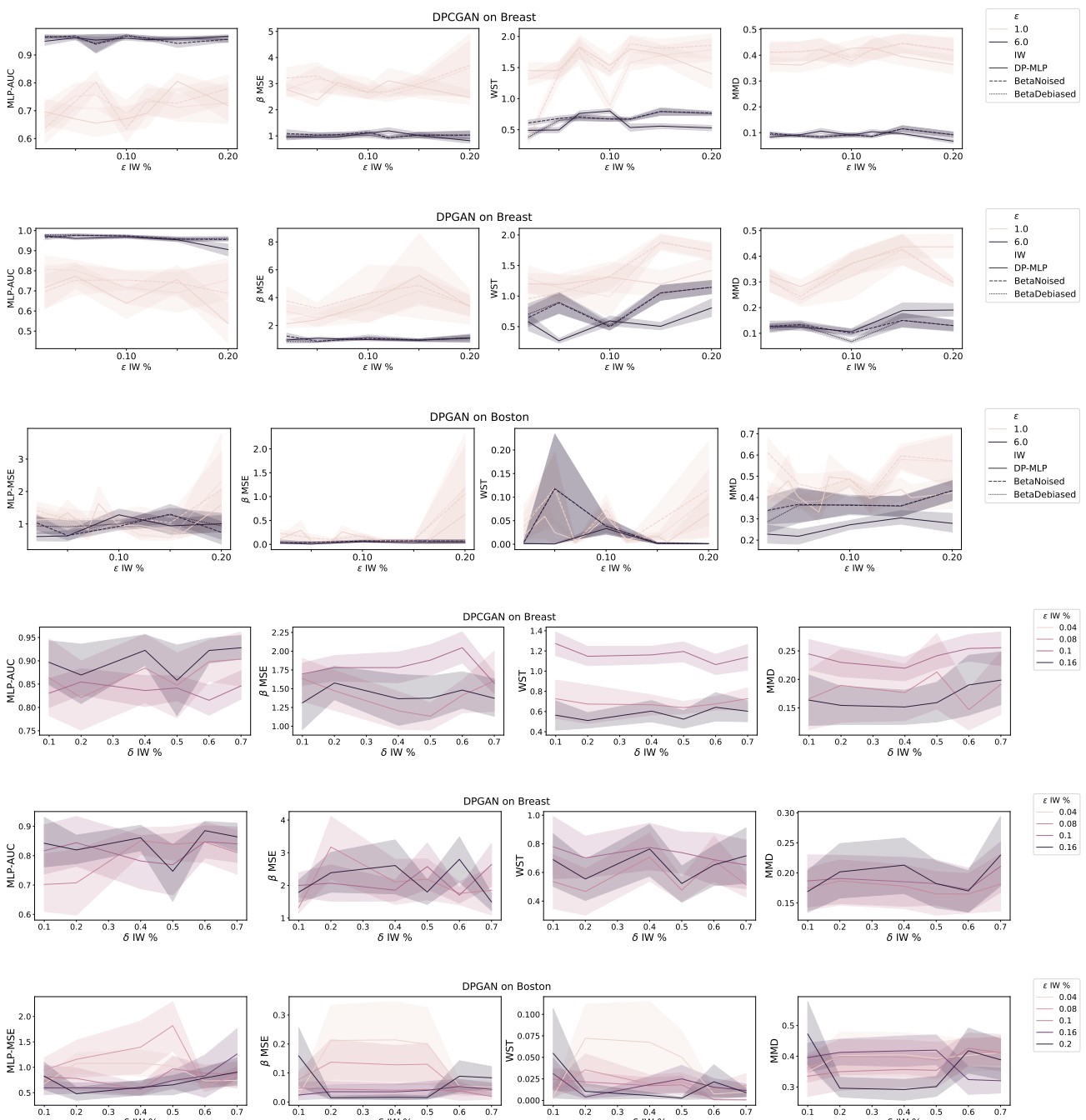

Figure 3: Multiple metrics measured across a range of privacy splits on Breast and Boston averaged over 10 seeds, and displayed with standard errors. The maximum mean discrepancy (MMD) was included as a measure of divergence between the weighted SDGP and the test distribution.

| SDGP | data | $\epsilon = 1$ | | $\epsilon = 6$ | |
|------|------|----------------|--|----------------|--|
| | | BetaNoised | BetaDebiased | BetaNoised | BetaDebiased |
| CGAN | Breast | $1.4833_{\pm 0.9603}$ | $\mathbf{0.0775_{\pm 0.0197}}$ | $0.0024_{\pm 0.0006}$ | $\mathbf{0.0020_{\pm 0.0004}}$ |
| | Banknote | $0.0420_{\pm 0.0211}$ | $\mathbf{0.0413_{\pm 0.0196}}$ | $\mathbf{0.0014_{\pm 0.0007}}$ | $\mathbf{0.0014_{\pm 0.0007}}$ |
| | Iris | $8.7522_{\pm 4.9893}$ | $\mathbf{3.4687_{\pm 1.3044}}$ | $\mathbf{0.1160_{\pm 0.0240}}$ | $0.1290_{\pm 0.0311}$ |
| GAN | Housing | $8.2081_{\pm 7.7702}$ | $\mathbf{1.4406_{\pm 0.8314}}$ | $3.7916_{\pm 3.3246}$ | $\mathbf{1.5479_{\pm 1.0430}}$ |
| DPCGAN | Breast | $0.0582_{\pm 0.0165}$ | $\mathbf{0.0445_{\pm 0.0162}}$ | $0.0015_{\pm 0.0003}$ | $\mathbf{0.0014_{\pm 0.0003}}$ |
| | Banknote | $0.0420_{\pm 0.0211}$ | $\mathbf{0.0413_{\pm 0.0196}}$ | $0.0022_{\pm 0.0013}$ | $\mathbf{0.0021_{\pm 0.0012}}$ |
| | Iris | $\mathbf{0.7834_{\pm 0.2341}}$ | $1.2300_{\pm 0.7050}$ | $\mathbf{0.2502_{\pm 0.1627}}$ | $0.2806_{\pm 0.1760}$ |
| DPGAN | Breast | $6.0487_{\pm 3.7927}$ | $\mathbf{3.7629_{\pm 2.2881}}$ | $0.0251_{\pm 0.0245}$ | $\mathbf{0.0238_{\pm 0.0234}}$ |
| | Banknote | $\mathbf{0.0582_{\pm 0.0353}}$ | $0.0610_{\pm 0.0397}$ | $0.0062_{\pm 0.0057}$ | $\mathbf{0.0061_{\pm 0.0056}}$ |
| | Iris | $2.6486_{\pm 1.3518}$ | $\mathbf{1.3698_{\pm 1.1554}}$ | $\mathbf{0.0741_{\pm 0.0228}}$ | $0.0864_{\pm 0.0274}$ |
| | Housing | $5.9175_{\pm 2.8546}$ | $\mathbf{0.8398_{\pm 0.6328}}$ | $\mathbf{1.9044_{\pm 1.1426}}$ | $2.1111_{\pm 1.3450}$ |

Table 6: Mean squared error averaged over 10 runs with standard errors reported in brackets for $(\epsilon = 1, \delta = 10^{-5})$ and $(\epsilon = 6, \delta = 10^{-5})$ where $\epsilon_{IW} = 0.1\epsilon$.

performance, significantly improving the performance measured by this metric, especially under synthetic data from the CGAN, DPCGAN and PrivBayes generators. Similar gains can be seen across the majority of the methods for the DPCGAN, especially at the higher $\epsilon = 6$.

All of these models were implemented in the `Turing.jl` PPL Ge et al. (2018). We then ran an experiment for each model and dataset on a defined grid across all seeds, synthetic generators and $\epsilon$ values. For each combination, we generated 10,000 samples across 4 chains (not counting 1,000 discarded warm-up samples per chain) for each of the importance weighting methods, as well as once for a model fit on the synthetic data with its standard non-weighted posterior, and once for the real data. We used Turing's implementation of the NUTS sampling algorithm with a target acceptance ratio of 0.65 for sampling the linear regression models' parameters, and for the logistic and multinomial logistic regression models we used HMC with a leapfrog step size of 0.05 and 10 leapfrog steps per iteration. The logistic and multinomial logistic regression models' coefficients (including intercepts) were given centred Normal priors with $\sigma = 1$. The linear regression models' coefficient priors were given the same centred Normal priors with $\sigma = 1$; its variance was given a non-informative prior via a truncated Normal distribution ensuring positivity with $\sigma = 10$.

We then took all 10,000 samples and calculated our evaluation metrics on the test set for each sample, storing all of these. We then present the distributions of metric scores that arise in the included box-plot figures.

## C.6   ILLUSTRATIVE EXAMPLE OF THE IMPLICATIONS OF BIAS MITIGATION

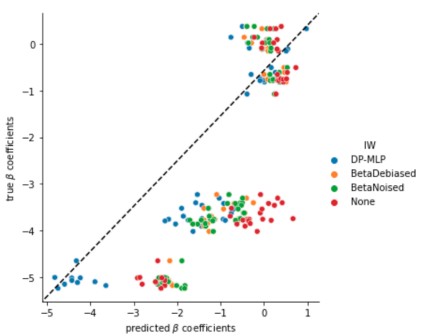

Figure 5: Illustrative example of debiasing with IW on PrivBayes synthesised Banknote data.

In Figure 5, we visualise the benefit of debiasing: We fitted a logistic regression as a downstream classifier on the private data to get the *true $\beta$ coefficients*. The *predicted $\beta$ coefficients* are estimated by training the logistic classifier on the importance weighted synthetic data. Each dot in the figure plots one dimension of the predicted $\beta$ coefficients against its true counterpart for one training run (out of ten). An optimal classifier would reconstruct the true coefficients. In this case all lines would be on the diagonal. An *unbiased* estimator would on average reconstruct the true coefficients: For each true $\beta$ coefficient, the predicted coefficients would be centred around the true value. We observe that coefficients learned without importance weighting exhibit the largest distance to the diagonal line, while the importance weighting alternatives push the dots closer to the diagonal line. Our method, DP-MLP, is particularly successful in decreasing the bias in the $\beta$ coefficients.

## C.7   COMPLETE UCI RESULTS

The complete experimental results on the UCI data sets can be found in Tables 7 to 10. Each table displays the performance of the different weight estimators for private and non-private synthetic data generative models for $\epsilon \in \{1, 6\}$, $\epsilon_{IW} = 0.1\epsilon$ and $\delta_{IW} = 0.3\delta$. We observe that importance weighting brings significant gains especially in low privacy regimes. For high privacy regimes this effect is reduced as the SDGP gets closer to the DGP.

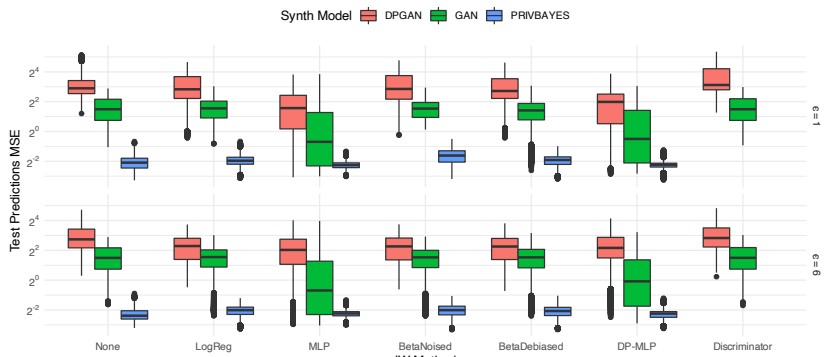

(a) Test set prediction MSE distributions calculated via chains of parameters sampled from a Bayesian linear regression model fit on synthesised TGFB data across 10 seeds.

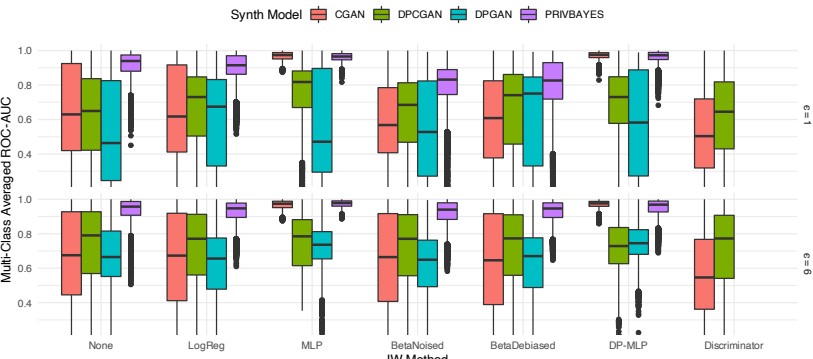

(b) Multi-class averaged ROC-AUC distributions calculated via chains of parameters sampled from a Bayesian multinomial logistic regression model fit on synthesised Iris data across 10 seeds.

| | | SDGP | CGAN | DPCGAN | DPGAN | PrivBayes |
|---|---|---|---|---|---|---|
| $\epsilon = 1$ | MLP-ROC-AUC $\uparrow$ | None | $0.4619_{\pm 0.1010}$ | $0.4717_{\pm 0.1103}$ | $0.5357_{\pm 0.0752}$ | $0.5243_{\pm 0.1299}$ |
| | | BetaNoised | $0.5824_{\pm 0.0931}$ | $0.5841_{\pm 0.0831}$ | $0.5487_{\pm 0.0803}$ | $\mathbf{0.6651_{\pm 0.0884}}$ |
| | | BetaDebiased | $0.5669_{\pm 0.1237}$ | $0.5913_{\pm 0.1136}$ | $0.5998_{\pm 0.1141}$ | $0.5005_{\pm 0.0793}$ |
| | | DP-MLP | $\mathbf{0.6299_{\pm 0.0984}}$ | $0.5725_{\pm 0.0859}$ | $0.5448_{\pm 0.0912}$ | $0.6143_{\pm 0.0374}$ |
| | | Discriminator | $0.5809_{\pm 0.0840}$ | $\mathbf{0.5995_{\pm 0.0982}}$ | $\mathbf{0.6475_{\pm 0.0701}}$ | - |
| | | LogReg | $0.4980_{\pm 0.0780}$ | $0.4908_{\pm 0.0950}$ | $0.4806_{\pm 0.0806}$ | $0.6245_{\pm 0.1235}$ |
| | | MLP | $0.7230_{\pm 0.0791}$ | $0.6273_{\pm 0.0988}$ | $0.5770_{\pm 0.1199}$ | $0.6778_{\pm 0.0923}$ |
| | $\beta$ MSE $\downarrow$ | None | $1.3594_{\pm 0.3789}$ | $1.0460_{\pm 0.2457}$ | $\mathbf{3.8955_{\pm 0.9764}}$ | $0.3511_{\pm 0.0753}$ |
| | | BetaNoised | $1.4944_{\pm 0.2321}$ | $1.1133_{\pm 0.1911}$ | $4.1565_{\pm 1.0469}$ | $0.4739_{\pm 0.0469}$ |
| | | BetaDebiased | $1.3682_{\pm 0.3080}$ | $1.3347_{\pm 0.2830}$ | $4.1694_{\pm 0.9246}$ | $0.8147_{\pm 0.1690}$ |
| | | DP-MLP | $\mathbf{0.6109_{\pm 0.0481}}$ | $1.0663_{\pm 0.1411}$ | $4.4986_{\pm 1.2881}$ | $\mathbf{0.1962_{\pm 0.0413}}$ |
| | | Discriminator | $1.0454_{\pm 0.3012}$ | $\mathbf{0.9404_{\pm 0.1024}}$ | $3.9049_{\pm 0.6010}$ | - |
| | | LogReg | $1.3345_{\pm 0.2725}$ | $0.9557_{\pm 0.1356}$ | $4.1971_{\pm 1.1035}$ | $0.3659_{\pm 0.0660}$ |
| | | MLP | $0.6091_{\pm 0.0546}$ | $0.8316_{\pm 0.1630}$ | $4.5109_{\pm 1.3057}$ | $0.1551_{\pm 0.0162}$ |
| | WST $\downarrow$ | None | $0.7226_{\pm 0.0543}$ | $0.7448_{\pm 0.0423}$ | $0.7919_{\pm 0.0458}$ | $0.5055_{\pm 0.0111}$ |
| | | BetaNoised | $0.2771_{\pm 0.0490}$ | $0.1014_{\pm 0.0519}$ | $0.1893_{\pm 0.0266}$ | $0.1412_{\pm 0.0493}$ |
| | | BetaDebiased | $\mathbf{0.2340_{\pm 0.0210}}$ | $\mathbf{0.0989_{\pm 0.0062}}$ | $0.1457_{\pm 0.0143}$ | $\mathbf{0.1059_{\pm 0.0032}}$ |
| | | DP-MLP | $0.3960_{\pm 0.0561}$ | $0.2376_{\pm 0.0196}$ | $0.2613_{\pm 0.0627}$ | $0.3451_{\pm 0.0253}$ |
| | | Discriminator | $0.2698_{\pm 0.0383}$ | $0.1696_{\pm 0.0371}$ | $\mathbf{0.1003_{\pm 0.0003}}$ | - |
| | | LogReg | $0.2341_{\pm 0.0687}$ | $0.1444_{\pm 0.0406}$ | $0.1611_{\pm 0.0178}$ | $0.3531_{\pm 0.0357}$ |
| | | MLP | $0.2677_{\pm 0.0693}$ | $0.0967_{\pm 0.0287}$ | $0.0752_{\pm 0.0261}$ | $0.1396_{\pm 0.0139}$ |
| $\epsilon = 6$ | MLP-ROC-AUC $\uparrow$ | None | $0.4662_{\pm 0.1039}$ | $0.5202_{\pm 0.0928}$ | $0.5252_{\pm 0.0844}$ | $0.4875_{\pm 0.1139}$ |
| | | BetaNoised | $0.5842_{\pm 0.0900}$ | $0.5531_{\pm 0.1093}$ | $0.5603_{\pm 0.0980}$ | $\mathbf{0.6218_{\pm 0.1304}}$ |
| | | BetaDebiased | $\mathbf{0.6029_{\pm 0.1100}}$ | $\mathbf{0.6992_{\pm 0.0801}}$ | $\mathbf{0.6445_{\pm 0.0906}}$ | $0.5388_{\pm 0.1258}$ |
| | | DP-MLP | $0.6007_{\pm 0.1060}$ | $0.6054_{\pm 0.0951}$ | $0.5181_{\pm 0.0957}$ | $0.5639_{\pm 0.0483}$ |
| | | Discriminator | $0.5894_{\pm 0.0829}$ | $0.5806_{\pm 0.1014}$ | $0.5909_{\pm 0.0903}$ | - |
| | | LogReg | $0.5073_{\pm 0.0852}$ | $0.5353_{\pm 0.0793}$ | $0.4934_{\pm 0.1051}$ | $0.7088_{\pm 0.0843}$ |
| | | MLP | $0.7206_{\pm 0.0774}$ | $0.7118_{\pm 0.0774}$ | $0.5923_{\pm 0.1130}$ | $0.6734_{\pm 0.0881}$ |
| | $\beta$ MSE $\downarrow$ | None | $1.4111_{\pm 0.3882}$ | $1.0262_{\pm 0.1866}$ | $\mathbf{2.0710_{\pm 0.3284}}$ | $0.2650_{\pm 0.0610}$ |
| | | BetaNoised | $1.2894_{\pm 0.2726}$ | $0.9507_{\pm 0.3017}$ | $2.8284_{\pm 1.0195}$ | $0.3338_{\pm 0.0701}$ |
| | | BetaDebiased | $1.2679_{\pm 0.2854}$ | $0.9511_{\pm 0.3113}$ | $2.8256_{\pm 1.0359}$ | $0.3492_{\pm 0.0719}$ |
| | | DP-MLP | $\mathbf{0.5928_{\pm 0.0682}}$ | $\mathbf{0.7773_{\pm 0.2286}}$ | $4.1112_{\pm 1.1372}$ | $\mathbf{0.2559_{\pm 0.0527}}$ |
| | | Discriminator | $1.0434_{\pm 0.3014}$ | $0.9449_{\pm 0.2838}$ | $2.1203_{\pm 0.5427}$ | - |
| | | LogReg | $1.2606_{\pm 0.2771}$ | $0.9604_{\pm 0.3155}$ | $2.8409_{\pm 1.0311}$ | $0.3603_{\pm 0.0806}$ |
| | | MLP | $0.6174_{\pm 0.0523}$ | $0.5102_{\pm 0.1630}$ | $3.9403_{\pm 1.1462}$ | $0.1283_{\pm 0.0252}$ |
| | WST $\downarrow$ | None | $0.7399_{\pm 0.0445}$ | $0.6598_{\pm 0.1077}$ | $0.6770_{\pm 0.0379}$ | $0.4255_{\pm 0.0208}$ |
| | | BetaNoised | $0.2703_{\pm 0.0492}$ | $0.3032_{\pm 0.0697}$ | $0.2622_{\pm 0.0229}$ | $0.4467_{\pm 0.0200}$ |
| | | BetaDebiased | $0.3035_{\pm 0.0601}$ | $0.3171_{\pm 0.0746}$ | $0.2770_{\pm 0.0332}$ | $\mathbf{0.3383_{\pm 0.0070}}$ |
| | | DP-MLP | $0.4507_{\pm 0.0722}$ | $0.5374_{\pm 0.0654}$ | $0.4445_{\pm 0.0635}$ | $0.4850_{\pm 0.0160}$ |
| | | Discriminator | $\mathbf{0.2134_{\pm 0.0419}}$ | $\mathbf{0.2168_{\pm 0.0032}}$ | $\mathbf{0.2178_{\pm 0.0037}}$ | - |
| | | LogReg | $0.3090_{\pm 0.0612}$ | $0.2836_{\pm 0.0742}$ | $0.2601_{\pm 0.0262}$ | $0.4591_{\pm 0.0121}$ |
| | | MLP | $0.2064_{\pm 0.0819}$ | $0.1343_{\pm 0.0299}$ | $0.2711_{\pm 0.0235}$ | $0.1981_{\pm 0.0192}$ |

Table 7: Results on Iris averaged over 10 seeds.

| $\epsilon$ | Metric | SDGP | CGAN | DPCGAN | DPGAN | PrivBayes |
|---|---|---|---|---|---|---|
| $\epsilon = 1$ | MLP-ROC-AUC $\leftarrow$ | None | $0.7408_{\pm 0.0522}$ | $0.8546_{\pm 0.0213}$ | $0.6863_{\pm 0.0436}$ | $0.7630_{\pm 0.0495}$ |
| | | BetaNoised | $0.7469_{\pm 0.0522}$ | $0.8495_{\pm 0.0274}$ | $0.6063_{\pm 0.0510}$ | $0.8943_{\pm 0.0173}$ |
| | | BetaDebiased | $\mathbf{0.7864_{\pm 0.0888}}$ | $\mathbf{0.8729_{\pm 0.0310}}$ | $0.5868_{\pm 0.1005}$ | $0.7632_{\pm 0.0517}$ |
| | | DP-MLP | $0.7313_{\pm 0.0613}$ | $0.7697_{\pm 0.0419}$ | $0.5657_{\pm 0.0570}$ | $\mathbf{0.8953_{\pm 0.0299}}$ |
| | | Discriminator | $0.7511_{\pm 0.0523}$ | $0.8695_{\pm 0.0167}$ | $\mathbf{0.7114_{\pm 0.0424}}$ | - |
| | | LogReg | $0.7986_{\pm 0.0391}$ | $0.8172_{\pm 0.0327}$ | $0.6034_{\pm 0.0534}$ | $0.9102_{\pm 0.0129}$ |
| | | MLP | $0.7253_{\pm 0.0521}$ | $0.8291_{\pm 0.0333}$ | $0.5974_{\pm 0.0627}$ | $0.8594_{\pm 0.0231}$ |
| | $\beta$ MSE $\downarrow$ | None | $15.3278_{\pm 2.5238}$ | $11.0215_{\pm 1.8377}$ | $39.3243_{\pm 3.7708}$ | $8.1724_{\pm 0.3987}$ |
| | | BetaNoised | $11.7636_{\pm 2.1960}$ | $8.4298_{\pm 1.0383}$ | $35.2862_{\pm 4.0365}$ | $5.7001_{\pm 0.1885}$ |
| | | BetaDebiased | $\mathbf{8.4946_{\pm 1.7858}}$ | $\mathbf{8.3508_{\pm 2.3127}}$ | $32.9909_{\pm 5.9024}$ | $6.6862_{\pm 0.1458}$ |
| | | DP-MLP | $14.6644_{\pm 2.9599}$ | $17.1597_{\pm 2.5448}$ | $36.4618_{\pm 4.1011}$ | $\mathbf{3.5519_{\pm 0.2895}}$ |
| | | Discriminator | $14.9537_{\pm 2.5553}$ | $12.5471_{\pm 2.3124}$ | $\mathbf{30.9282_{\pm 5.4283}}$ | - |
| | | LogReg | $11.7777_{\pm 2.2000}$ | $8.4760_{\pm 1.0406}$ | $35.2964_{\pm 4.0396}$ | $5.6751_{\pm 0.1785}$ |
| | | MLP | $15.4584_{\pm 3.0826}$ | $17.9390_{\pm 2.4926}$ | $35.5211_{\pm 4.2147}$ | $2.6286_{\pm 0.3761}$ |
| | WST $\rightarrow$ | None | $0.6702_{\pm 0.0282}$ | $0.4746_{\pm 0.0214}$ | $0.7442_{\pm 0.0333}$ | $0.3237_{\pm 0.0162}$ |
| | | BetaNoised | $0.3106_{\pm 0.0475}$ | $0.2509_{\pm 0.0436}$ | $0.4355_{\pm 0.0456}$ | $0.2318_{\pm 0.0035}$ |
| | | BetaDebiased | $0.3837_{\pm 0.0990}$ | $0.4015_{\pm 0.0766}$ | $0.4618_{\pm 0.0832}$ | $0.2369_{\pm 0.0061}$ |
| | | DP-MLP | $\mathbf{0.1418_{\pm 0.0283}}$ | $\mathbf{0.2035_{\pm 0.0427}}$ | $0.4298_{\pm 0.0433}$ | $\mathbf{0.0456_{\pm 0.0061}}$ |
| | | Discriminator | $0.6366_{\pm 0.0273}$ | $0.3382_{\pm 0.0399}$ | $\mathbf{0.1087_{\pm 0.0415}}$ | - |
| | | LogReg | $0.3092_{\pm 0.0470}$ | $0.2508_{\pm 0.0432}$ | $0.4348_{\pm 0.0460}$ | $0.2348_{\pm 0.0034}$ |
| | | MLP | $0.0494_{\pm 0.0141}$ | $0.0913_{\pm 0.0259}$ | $0.3860_{\pm 0.0452}$ | $0.0021_{\pm 0.0004}$ |
| $\epsilon = 6$ | MLP-ROC-AUC $\leftarrow$ | None | $0.7212_{\pm 0.0491}$ | $0.8958_{\pm 0.0179}$ | $\mathbf{0.8323_{\pm 0.0301}}$ | $0.8357_{\pm 0.0354}$ |
| | | BetaNoised | $\mathbf{0.7811_{\pm 0.0423}}$ | $0.8771_{\pm 0.0227}$ | $0.8216_{\pm 0.0320}$ | $0.8588_{\pm 0.0295}$ |
| | | BetaDebiased | $0.6951_{\pm 0.0958}$ | $\mathbf{0.8992_{\pm 0.0334}}$ | $0.7061_{\pm 0.1083}$ | $0.8136_{\pm 0.0648}$ |
| | | DP-MLP | $0.6879_{\pm 0.0547}$ | $0.8582_{\pm 0.0330}$ | $0.7445_{\pm 0.0511}$ | $\mathbf{0.8899_{\pm 0.0148}}$ |
| | | Discriminator | $0.7332_{\pm 0.0529}$ | $0.8976_{\pm 0.0148}$ | $0.8071_{\pm 0.0362}$ | - |
| | | LogReg | $0.7953_{\pm 0.0421}$ | $0.8867_{\pm 0.0207}$ | $0.7871_{\pm 0.0351}$ | $0.8668_{\pm 0.0336}$ |
| | | MLP | $0.6960_{\pm 0.0456}$ | $0.8599_{\pm 0.0291}$ | $0.8025_{\pm 0.0212}$ | $0.8404_{\pm 0.0400}$ |
| | $\beta$ MSE $\downarrow$ | None | $19.2959_{\pm 4.0480}$ | $8.3074_{\pm 1.6718}$ | $18.0835_{\pm 2.5051}$ | $7.9052_{\pm 0.3837}$ |
| | | BetaNoised | $14.4350_{\pm 2.3116}$ | $6.4683_{\pm 0.9572}$ | $23.0590_{\pm 3.2307}$ | $5.4736_{\pm 0.1792}$ |
| | | BetaDebiased | $\mathbf{13.1578_{\pm 2.9727}}$ | $\mathbf{5.6890_{\pm 1.0695}}$ | $19.1627_{\pm 6.1430}$ | $6.4776_{\pm 0.1134}$ |
| | | DP-MLP | $18.7059_{\pm 3.0658}$ | $8.8820_{\pm 1.4421}$ | $24.0433_{\pm 3.4451}$ | $\mathbf{3.0883_{\pm 0.2703}}$ |
| | | Discriminator | $18.9194_{\pm 4.0483}$ | $8.0682_{\pm 1.5928}$ | $\mathbf{13.6267_{\pm 1.9313}}$ | - |
| | | LogReg | $14.4464_{\pm 2.3126}$ | $6.4701_{\pm 0.9581}$ | $23.0696_{\pm 3.2327}$ | $5.4706_{\pm 0.1781}$ |
| | | MLP | $18.2400_{\pm 3.1143}$ | $9.7111_{\pm 1.4901}$ | $23.0268_{\pm 3.2550}$ | $2.4589_{\pm 0.3184}$ |
| | WST $\rightarrow$ | None | $0.6642_{\pm 0.0270}$ | $0.4723_{\pm 0.0294}$ | $0.5645_{\pm 0.0219}$ | $0.2928_{\pm 0.0118}$ |
| | | BetaNoised | $0.2507_{\pm 0.0384}$ | $0.3078_{\pm 0.0231}$ | $0.2608_{\pm 0.0370}$ | $0.2269_{\pm 0.0036}$ |
| | | BetaDebiased | $0.2316_{\pm 0.0670}$ | $0.2892_{\pm 0.0442}$ | $0.3029_{\pm 0.0883}$ | $0.2176_{\pm 0.0076}$ |
| | | DP-MLP | $\mathbf{0.1395_{\pm 0.0262}}$ | $\mathbf{0.0957_{\pm 0.0183}}$ | $0.1730_{\pm 0.0413}$ | $\mathbf{0.1142_{\pm 0.0017}}$ |
| | | Discriminator | $0.6303_{\pm 0.0278}$ | $0.3596_{\pm 0.0470}$ | $\mathbf{0.0436_{\pm 0.0100}}$ | - |
| | | LogReg | $0.2504_{\pm 0.0384}$ | $0.3083_{\pm 0.0231}$ | $0.2607_{\pm 0.0370}$ | $0.2272_{\pm 0.0035}$ |
| | | MLP | $0.0658_{\pm 0.0208}$ | $0.0409_{\pm 0.0104}$ | $0.0787_{\pm 0.0325}$ | $0.2025_{\pm 0.0004}$ |

Table 8: Results on Banknote averaged over 10 seeds.

| | | SDGP | GAN | DPGAN | PrivBayes |
|---|---|---|---|---|---|
| $\epsilon = 1$ | MLP MSE $\rightarrow$ | None | $1.4464_{\pm 0.1591}$ | $1.8851_{\pm 0.5262}$ | $0.1973_{\pm 0.0108}$ |
| | | BetaNoised | $0.6455_{\pm 0.0942}$ | $1.0057_{\pm 0.1973}$ | $0.2200_{\pm 0.0154}$ |
| | | BetaDebiased | $\mathbf{0.6421_{\pm 0.1290}}$ | $\mathbf{0.9024_{\pm 0.1244}}$ | $0.2139_{\pm 0.0122}$ |
| | | DP-MLP | $0.8279_{\pm 0.0974}$ | $0.9462_{\pm 0.1702}$ | $\mathbf{0.1877_{\pm 0.0174}}$ |
| | | Discriminator | $1.5126_{\pm 0.1639}$ | $1.6256_{\pm 0.2394}$ | - |
| | | LogReg | $0.6292_{\pm 0.0909}$ | $1.0606_{\pm 0.2648}$ | $0.2515_{\pm 0.0305}$ |
| | | MLP | $0.6266_{\pm 0.1273}$ | $1.0979_{\pm 0.2225}$ | $0.1697_{\pm 0.0079}$ |
| | $\beta$ MSE $\downarrow$ | None | $0.1017_{\pm 0.0118}$ | $0.1867_{\pm 0.0434}$ | $\mathbf{0.0011_{\pm 0.0002}}$ |
| | | BetaNoised | $0.0601_{\pm 0.0172}$ | $0.1761_{\pm 0.0948}$ | $0.0088_{\pm 0.0028}$ |
| | | BetaDebiased | $0.0608_{\pm 0.0190}$ | $\mathbf{0.0667_{\pm 0.0188}}$ | $0.0077_{\pm 0.0022}$ |
| | | DP-MLP | $\mathbf{0.0363_{\pm 0.0192}}$ | $0.1530_{\pm 0.0812}$ | $0.0048_{\pm 0.0024}$ |
| | | Discriminator | $0.0940_{\pm 0.0100}$ | $0.1567_{\pm 0.1825}$ | - |
| | | LogReg | $0.0707_{\pm 0.0194}$ | $0.0749_{\pm 0.0279}$ | $0.0037_{\pm 0.0016}$ |
| | | MLP | $0.0058_{\pm 0.0007}$ | $0.1476_{\pm 0.0804}$ | $0.0008_{\pm 0.0002}$ |
| | WST $\rightarrow$ | None | $1.3060_{\pm 0.0319}$ | $2.2013_{\pm 0.0945}$ | $1.3938_{\pm 0.0231}$ |
| | | BetaNoised | $1.0060_{\pm 0.0023}$ | $2.0922_{\pm 0.0419}$ | $1.3009_{\pm 0.0338}$ |
| | | BetaDebiased | $1.0023_{\pm 0.0009}$ | $2.0930_{\pm 0.0393}$ | $1.2705_{\pm 0.0290}$ |
| | | DP-MLP | $1.0036_{\pm 0.0015}$ | $2.0542_{\pm 0.0184}$ | $\mathbf{1.0265_{\pm 0.0035}}$ |
| | | Discriminator | $\mathbf{0.9472_{\pm 0.0764}}$ | $\mathbf{2.0145_{\pm 0.0141}}$ | - |
| | | LogReg | $1.0070_{\pm 0.0042}$ | $2.2051_{\pm 0.0819}$ | $1.4078_{\pm 0.0492}$ |
| | | MLP | $1.0001_{\pm 0.0001}$ | $2.0350_{\pm 0.0158}$ | $1.0072_{\pm 0.0009}$ |
| $\epsilon = 6$ | MLP MSE $\rightarrow$ | None | $1.8218_{\pm 0.1514}$ | $1.8016_{\pm 0.1771}$ | $0.1633_{\pm 0.0074}$ |
| | | BetaNoised | $\mathbf{0.5318_{\pm 0.0806}}$ | $\mathbf{0.6529_{\pm 0.0814}}$ | $0.1940_{\pm 0.0156}$ |
| | | BetaDebiased | $0.5647_{\pm 0.1065}$ | $0.9025_{\pm 0.1462}$ | $0.1810_{\pm 0.0131}$ |
| | | DP-MLP | $0.9737_{\pm 0.1178}$ | $1.0902_{\pm 0.1486}$ | $\mathbf{0.1428_{\pm 0.0068}}$ |
| | | Discriminator | $1.8398_{\pm 0.1446}$ | $1.8631_{\pm 0.1986}$ | - |
| | | LogReg | $0.5501_{\pm 0.0540}$ | $0.9050_{\pm 0.1553}$ | $0.1934_{\pm 0.0224}$ |
| | | MLP | $0.4725_{\pm 0.0736}$ | $0.7464_{\pm 0.1185}$ | $0.1581_{\pm 0.0076}$ |
| | $\beta$ MSE $\downarrow$ | None | $0.1230_{\pm 0.0110}$ | $0.1450_{\pm 0.0174}$ | $0.0009_{\pm 0.0002}$ |
| | | BetaNoised | $0.0695_{\pm 0.0203}$ | $0.0608_{\pm 0.0231}$ | $0.0022_{\pm 0.0006}$ |
| | | BetaDebiased | $0.0693_{\pm 0.0207}$ | $0.0613_{\pm 0.0240}$ | $0.0018_{\pm 0.0004}$ |
| | | DP-MLP | $\mathbf{0.0030_{\pm 0.0006}}$ | $\mathbf{0.0354_{\pm 0.0112}}$ | $\mathbf{0.0008_{\pm 0.0002}}$ |
| | | Discriminator | $0.1135_{\pm 0.0098}$ | $0.2274_{\pm 0.0375}$ | - |
| | | LogReg | $0.0697_{\pm 0.0207}$ | $0.0606_{\pm 0.0237}$ | $0.0018_{\pm 0.0004}$ |
| | | MLP | $0.0063_{\pm 0.0011}$ | $0.0212_{\pm 0.0060}$ | $0.0008_{\pm 0.0001}$ |
| | WST $\rightarrow$ | None | $1.3727_{\pm 0.0249}$ | $1.5681_{\pm 0.0368}$ | $1.3306_{\pm 0.0271}$ |
| | | BetaNoised | $1.0031_{\pm 0.0012}$ | $1.0615_{\pm 0.0304}$ | $1.3906_{\pm 0.0410}$ |
| | | BetaDebiased | $\mathbf{1.0031_{\pm 0.0012}}$ | $1.0598_{\pm 0.0286}$ | $1.4106_{\pm 0.0432}$ |
| | | DP-MLP | $1.0140_{\pm 0.0032}$ | $\mathbf{1.0338_{\pm 0.0126}}$ | $\mathbf{1.2405_{\pm 0.0133}}$ |
| | | Discriminator | $1.0481_{\pm 0.0752}$ | $1.3844_{\pm 0.0654}$ | - |
| | | LogReg | $1.0031_{\pm 0.0012}$ | $1.0623_{\pm 0.0298}$ | $1.4033_{\pm 0.0406}$ |
| | | MLP | $1.0001_{\pm 0.0000}$ | $1.0081_{\pm 0.0045}$ | $1.0097_{\pm 0.0010}$ |

Table 9: Results on Boston averaged over 10 seeds.

| | | SDGP | CGAN | DPCGAN | DPGAN | PrivBayes |
|---|---|---|---|---|---|---|
| $\epsilon=1$ | MLP-ROC-AUC ← | None | $0.6801_{\pm0.0655}$ | $0.6374_{\pm0.0421}$ | $0.6791_{\pm0.0966}$ | $0.8366_{\pm0.0579}$ |
| | | BetaNoised | $0.7732_{\pm0.0589}$ | $0.6110_{\pm0.0477}$ | $0.6546_{\pm0.0727}$ | $0.7076_{\pm0.0983}$ |
| | | BetaDebiased | $0.7151_{\pm0.1146}$ | $0.6820_{\pm0.0510}$ | $0.7173_{\pm0.0842}$ | $\mathbf{0.8557_{\pm0.0765}}$ |
| | | DP-MLP | $0.7166_{\pm0.1038}$ | $\mathbf{0.7942_{\pm0.0404}}$ | $0.5686_{\pm0.0823}$ | $0.7353_{\pm0.0887}$ |
| | | Discriminator | $\mathbf{0.8607_{\pm0.0485}}$ | $0.6992_{\pm0.0839}$ | $\mathbf{0.7290_{\pm0.0720}}$ | - |
| | | LogReg | $0.7141_{\pm0.0755}$ | $0.6631_{\pm0.0469}$ | $0.6484_{\pm0.1081}$ | $0.7618_{\pm0.1019}$ |
| | | MLP | $0.6942_{\pm0.1262}$ | $0.7730_{\pm0.0412}$ | $0.7358_{\pm0.1017}$ | $0.7573_{\pm0.0738}$ |
| | $\beta$ MSE ↓ | None | $2.3646_{\pm0.2983}$ | $2.0643_{\pm0.2012}$ | $4.9828_{\pm1.5701}$ | $2.3904_{\pm0.1050}$ |
| | | BetaNoised | $1.4900_{\pm0.1807}$ | $2.7532_{\pm0.2650}$ | $2.5025_{\pm0.3763}$ | $2.1144_{\pm0.2400}$ |
| | | BetaDebiased | $1.5413_{\pm0.2378}$ | $2.8337_{\pm0.3842}$ | $\mathbf{2.2324_{\pm1.0446}}$ | $\mathbf{1.8266_{\pm0.2392}}$ |
| | | DP-MLP | $\mathbf{0.9977_{\pm0.1617}}$ | $2.3965_{\pm0.2083}$ | $3.8865_{\pm0.6043}$ | $2.3130_{\pm0.2195}$ |
| | | Discriminator | $1.8554_{\pm0.3263}$ | $\mathbf{1.4591_{\pm0.1837}}$ | $4.0612_{\pm0.9523}$ | - |
| | | LogReg | $1.1940_{\pm0.1610}$ | $2.6934_{\pm0.2667}$ | $2.2156_{\pm0.3366}$ | $1.5333_{\pm0.2138}$ |
| | | MLP | $1.0120_{\pm0.1383}$ | $2.3999_{\pm0.2040}$ | $3.8343_{\pm0.7032}$ | $1.6581_{\pm0.2020}$ |
| | WST ↓ | None | $1.8426_{\pm0.1329}$ | $2.3665_{\pm0.0982}$ | $1.5853_{\pm0.1333}$ | $2.1117_{\pm0.1740}$ |
| | | BetaNoised | $1.3109_{\pm0.0507}$ | $1.4337_{\pm0.1114}$ | $2.2232_{\pm0.2325}$ | $1.2322_{\pm0.0823}$ |
| | | BetaDebiased | $\mathbf{1.0649_{\pm0.0120}}$ | $1.8922_{\pm0.1237}$ | $1.9913_{\pm0.3507}$ | $\mathbf{1.1825_{\pm0.0933}}$ |
| | | DP-MLP | $1.4737_{\pm0.1027}$ | $1.4570_{\pm0.1492}$ | $1.0315_{\pm0.1415}$ | $1.2190_{\pm0.0795}$ |
| | | Discriminator | $1.8814_{\pm0.1682}$ | $\mathbf{1.0007_{\pm0.0004}}$ | $\mathbf{1.0001_{\pm0.0001}}$ | - |
| | | LogReg | $1.4374_{\pm0.0467}$ | $1.6451_{\pm0.1168}$ | $2.2953_{\pm0.2121}$ | $1.4663_{\pm0.1152}$ |
| | | MLP | $1.3056_{\pm0.0524}$ | $1.6129_{\pm0.1404}$ | $1.0709_{\pm0.1579}$ | $1.4141_{\pm0.1216}$ |
| $\epsilon=6$ | MLP-ROC-AUC ← | None | $0.6177_{\pm0.0737}$ | $\mathbf{0.9790_{\pm0.0058}}$ | $\mathbf{0.9756_{\pm0.0042}}$ | $0.9435_{\pm0.0152}$ |
| | | BetaNoised | $0.7185_{\pm0.0898}$ | $0.9715_{\pm0.0031}$ | $0.9710_{\pm0.0065}$ | $0.9699_{\pm0.0121}$ |
| | | BetaDebiased | $\mathbf{0.9070_{\pm0.0434}}$ | $0.9723_{\pm0.0033}$ | $0.9724_{\pm0.0066}$ | $\mathbf{0.9820_{\pm0.0064}}$ |
| | | DP-MLP | $0.7203_{\pm0.1028}$ | $0.9703_{\pm0.0040}$ | $0.9728_{\pm0.0059}$ | $0.9754_{\pm0.0063}$ |
| | | Discriminator | $0.8712_{\pm0.0471}$ | $0.9763_{\pm0.0071}$ | $0.9737_{\pm0.0065}$ | - |
| | | LogReg | $0.6869_{\pm0.0760}$ | $0.9706_{\pm0.0033}$ | $0.9719_{\pm0.0049}$ | $0.9825_{\pm0.0061}$ |
| | | MLP | $0.6899_{\pm0.1290}$ | $0.9584_{\pm0.0080}$ | $0.9767_{\pm0.0043}$ | $0.9506_{\pm0.0250}$ |
| | $\beta$ MSE ↓ | None | $2.3602_{\pm0.4035}$ | $0.9886_{\pm0.2287}$ | $1.0653_{\pm0.1229}$ | $\mathbf{0.9142_{\pm0.1575}}$ |
| | | BetaNoised | $1.2400_{\pm0.1637}$ | $1.0329_{\pm0.0732}$ | $1.1586_{\pm0.1312}$ | $1.0465_{\pm0.1358}$ |
| | | BetaDebiased | $\mathbf{0.9388_{\pm0.0802}}$ | $1.0150_{\pm0.0783}$ | $1.1617_{\pm0.1936}$ | $0.9843_{\pm0.1766}$ |
| | | DP-MLP | $0.9949_{\pm0.1486}$ | $1.0119_{\pm0.0698}$ | $0.8969_{\pm0.0837}$ | $1.3442_{\pm0.0900}$ |
| | | Discriminator | $1.7588_{\pm0.3421}$ | $\mathbf{0.8539_{\pm0.2323}}$ | $\mathbf{0.5423_{\pm0.0457}}$ | - |
| | | LogReg | $1.2221_{\pm0.1598}$ | $1.0310_{\pm0.0719}$ | $1.1484_{\pm0.1276}$ | $1.0234_{\pm0.1274}$ |
| | | MLP | $1.0845_{\pm0.1210}$ | $1.0953_{\pm0.0844}$ | $0.9275_{\pm0.0938}$ | $1.5354_{\pm0.1343}$ |
| | WST ↓ | None | $1.8436_{\pm0.1257}$ | $1.3378_{\pm0.0282}$ | $1.6449_{\pm0.0849}$ | $2.0437_{\pm0.2188}$ |
| | | BetaNoised | $1.4164_{\pm0.0483}$ | $0.6526_{\pm0.0463}$ | $1.5485_{\pm0.0635}$ | $1.4808_{\pm0.0943}$ |
| | | BetaDebiased | $\mathbf{1.3314_{\pm0.0459}}$ | $0.6641_{\pm0.0482}$ | $1.5156_{\pm0.0935}$ | $\mathbf{1.4133_{\pm0.1346}}$ |
| | | DP-MLP | $1.7176_{\pm0.1206}$ | $0.7931_{\pm0.0380}$ | $1.5551_{\pm0.0826}$ | $1.4923_{\pm0.0685}$ |
| | | Discriminator | $1.8523_{\pm0.1553}$ | $\mathbf{0.2363_{\pm0.0425}}$ | $\mathbf{1.1020_{\pm0.0158}}$ | - |
| | | LogReg | $1.4140_{\pm0.0493}$ | $0.6597_{\pm0.0470}$ | $1.5281_{\pm0.0622}$ | $1.4824_{\pm0.0952}$ |
| | | MLP | $1.3487_{\pm0.0591}$ | $0.3762_{\pm0.0383}$ | $1.2309_{\pm0.0387}$ | $1.3406_{\pm0.0792}$ |

Table 10: Results on Breast averaged over 10 seeds.

## C.8 COMPARISON TO EXPERIMENTAL RESULTS REPORTED BY RELATED WORK

We compare our results to PATE-GAN and DPGAN as DP synthetic data generators (Jordon et al., 2019; Xie et al., 2018). The PATEGAN implementation is taken from `https://github.com/vanderschaarlab/mlforhealthlabpub`. For DPGAN we chose the code from the DataSynthesizer package. In the implementation of the PATE-GAN method, Jordon et al. (2019) generate 50 independent synthetic data sets for each function call, returning the best synthetic data set as defined by a comparison with non-private validation data. The relative level of privacy violation in these situations is unknown, making interpretation of results and comparison between methods in tables and figures challenging. On re-implementing the methods to generate DP synthetic data, we find a substantial and significant drop in performance, which nonetheless is improved through bias mitigation. Please see the GitHub repository for further results and an illustration why PATE GAN underperforms.