# OpenReview forum: "Mitigating Statistical Bias within Differentially Private Synthetic Data"
_auai.org/UAI/2022/Conference — UAI 2022 Oral_

### Official Review · Reviewer_YjUv · 2022-03-25

**Q2(1) Originality/Novelty:** 3
**Q2(2) Significance/Impact:** 2
**Q2(3) Correctness/Technical Quality:** 3
**Q2(6) Clarity Of Writing:** 3
**Q6 Overall Score:** 6
**Q8 Confidence In Your Score:** 3

**Q1 Summary And Contributions:**

This paper considers the problem of augmenting synthetic data with importance weights to improve downstream tasks, all in the context of differential privacy (DP). They empirically compares several approaches to creating DP importance weights (IWs). On the technical side, they show that an earlier method of getting DP IWs leads to biased importance sampling estimators, and they provide a new adjusted importance weight that removes this bias and decreases variance.


**Q2 Assessment Of The Paper:**

More detailed information regarding each of these aspects is given below:

**Q2(4) Quality Of Experiments (Optional):**

3: Good: The experimental evaluation is adequate, and the results convincingly support the main claims.

**Q2(5) Reproducibility:**

4: Excellent: Key resources (e.g., proofs, code, data) are available and key details (e.g., proof sketches, experimental setup) are comprehensively described for competent researchers to confidently and easily reproduce the main results.

**Q3 Main Strengths:**

- Contributes a way to adjust logistic regression based importance weights so that downstream importance sampling estimates are unbiased.
- Ran a fairly large number of experiments comparing performance of 3 synthetic data generation methods together with several different approaches to generating importance weights, on 6 different datasets and 3 performance metrics.
- Seems to offer strong evidence for the advantages of some kind of importance weighting.


**Q4 Main Weakness:**

- The performance of various importance weight methods vary widely,
and it's not clear what's driving it. In general, lots of experiments, but few clear takeaways.


**Q5 Detailed Comments To The Authors:**

- In section 2.2, you say "If Di is (eps,delta)-DP, then any composition $Di \circ Ge$ is also (eps, delta)-DP, since Ge does not query the protected data."  You've defined Di as a function on an input space X, rather than on a dataset, but as such, the definition of differential privacy doesn't apply directly to it.  Might be worth clarifying.
- In Proposition 3,  the claim is about differential privacy w.r.t. the full dataset -- this seems strange when comparing to the Abadi paper where it seems things are framed in terms of the number of steps of SGD.  By w.r.t. the full dataset, are you really saying for a number of steps T = (N_D + N_G) / L equivalent to one pass through the data?  Or am I wrong that we keep using more privacy budget as we take more steps, say for more multiple epochs?  Some clarification here would be helpful.


**Q7 Justification For Your Score:**

I think the adjusted importance weights are a good theoretical contribution, and I think the experiments are more than sufficient to accompany that contribution.  My score would be higher if the empirical work had a more clear message, though I'm not sure how to get there.

**Q9 Complying With Reviewing Instructions:**

1: Yes.

---

### Official Review · Reviewer_oWTq · 2022-03-27

**Q2(1) Originality/Novelty:** 3
**Q2(2) Significance/Impact:** 3
**Q2(3) Correctness/Technical Quality:** 3
**Q2(6) Clarity Of Writing:** 4
**Q6 Overall Score:** 7
**Q8 Confidence In Your Score:** 4

**Q1 Summary And Contributions:**

The paper proposes using importance weights to mitigate bias in downstream tasks using synthetic data generated by differentially private generative models. The weights are simply the ratios of the densities of the real data and the generator. A general framework is provided for incorporating the weights in downstream tasks and a logistic regression and general method where SGD is used to train an MLP are provided for learning weights. Experiments demonstrate downstream performance gains.

**Q2 Assessment Of The Paper:**

More detailed information regarding each of these aspects is given below:

**Q2(4) Quality Of Experiments (Optional):**

3: Good: The experimental evaluation is adequate, and the results convincingly support the main claims.

**Q2(5) Reproducibility:**

3: Good: Key resources (e.g., proofs, code, data) are available and key details (e.g., proofs, experimental setup) are sufficiently well-described for competent researchers to confidently reproduce the main results.

**Q3 Main Strengths:**

- DP generative models are known to suffer from poor fidelity at reasonable privacy levels. This approach helps improve fidelity while maintaining privacy levels.

- It is generic enough that it can be applied to different types of DP generative models.

- The proposed "privatized neural network" approach for learning weights is flexible and can be easily adapted to other architectures or models.

- Extensive experiments demonstrate performance gains on downstream tasks

**Q4 Main Weakness:**

- The approach is not end-to-end, resulting in unbiased samples, but rather samples + IWs which downstream tasks must incorporate

- While the paper is well written and generally clear, it is not sufficiently clear on whether/when an additional privacy budget is required for estimating IW

- Some useful baselines are not included

**Q5 Detailed Comments To The Authors:**

- My primary uncertainty is in regards to whether and when an additional privacy budget is required for estimating the IW. In 3.5, the paper indicates that an additional budget is not required if a GAN is used and the weights are computed from the discriminator, appealing to the post-processing theorem because the discriminator parameters are already DP. However, if I understand correctly, the discriminator still needs to be applied to the real data (as in algorithm 1) so this would not negate the need for an additional privacy budget. 3.5 indicates that more details on why an additional privacy budget is not required are indicated in section 4, but I cannot find these details.

- The paper cites PATE in several times, but does not include PATE in any experiments? This seems odd since PATE generally outperforms DPGAN. Why is it excluded? Have any experiments been performed using PATE?

- Requiring that downstream procedures incorporate biased synthetic data + IWs is less natural than simply modifying the generative procedure / reweighting scheme to be end-to-end resulting in unbiased synthetic data. Have the authors considered whether the same ideas could be incorporated to directly produce unbiased synthetic data?

**Q7 Justification For Your Score:**

The approach addresses a relevant issue in privacy-preserving ML. The solution is flexible; it can be applied to multiple DP generative models. It is well justified theoretically and empirically.

**Q9 Complying With Reviewing Instructions:**

1: Yes.

---

### Official Review · Reviewer_q3jE · 2022-04-13

**Q2(1) Originality/Novelty:** 3
**Q2(2) Significance/Impact:** 3
**Q2(3) Correctness/Technical Quality:** 3
**Q2(6) Clarity Of Writing:** 3
**Q6 Overall Score:** 7
**Q8 Confidence In Your Score:** 3

**Q1 Summary And Contributions:**

The main contribution of the paper is the proposal and study of several alternatives for using importance weighting to reduce bias in differential privacy. The methods introduced provide reasonably good (although not groundbreaking) results on the experimental analysis.

**Q10 Ethical Concerns (Optional):**

No ethical concerns

**Q2 Assessment Of The Paper:**

More detailed information regarding each of these aspects is given below:

**Q2(4) Quality Of Experiments (Optional):**

3: Good: The experimental evaluation is adequate, and the results convincingly support the main claims.

**Q2(5) Reproducibility:**

4: Excellent: Key resources (e.g., proofs, code, data) are available and key details (e.g., proof sketches, experimental setup) are comprehensively described for competent researchers to confidently and easily reproduce the main results.

**Q3 Main Strengths:**

The paper properly acknowledges existing work and clearly presents the new results.

Unbiased estimation from differentially private data is a very rellevant problem in AI.

The paper looks technically very sound, and the experiments seem well designed, but I have not checked the proofs in detail.

Full access to the code is promised and the experimental description is very detailed, making it easy to reproduce the results.

**Q4 Main Weakness:**

No major weaknesses identified

**Q5 Detailed Comments To The Authors:**

As a minor weakness, the presentation of the results and conclusions should be easier to follow. I found it a bit difficult to trace the main contributions described in the introduction to the different sections of the paper.
The presentation of  the evaluation metrics in page 7 is too concise and a bit difficult to understand. Specifically the sentence
"As an exemplary supervised downstream task, we consider the training of a linear downstream classifier or regressor on
the synthetic data which is assessed by the area under the receiver operating curve (ROC-AUC) or the MSE on the private test dataset (MLP ROC-AUC and MLP MSE)."

Are MLP ROC-AUC and MLP MSE coming from a linear model of from an MLP?

**Q7 Justification For Your Score:**

The paper presents two methods for using importance weighting to debias inferences based on differentially private data. The framework for presenting and analyzing the methods is well thought out. The methods provide reasonably good results. I think the papers has the quality and novelty required for UAI.

**Q9 Complying With Reviewing Instructions:**

1: Yes.

---

### Official Review · Reviewer_tBHj · 2022-04-13

**Q2(1) Originality/Novelty:** 3
**Q2(2) Significance/Impact:** 3
**Q2(3) Correctness/Technical Quality:** 4
**Q2(6) Clarity Of Writing:** 4
**Q6 Overall Score:** 7
**Q8 Confidence In Your Score:** 3

**Q1 Summary And Contributions:**

The paper consider importance sampling approach to correct the bias in synthetic data obtained with a differentially private algorithm. Importance sampling for synthetic data has been considered before in the non-DP setting in the literature mentioned in the paper ( Grover et al. (2019) and so on), however not really addressed before in the non-DP setting. Having bit of public data, the DP synthetic data can be importance sampled and as a result the bias of downstream estimators is mitigated.

**Q2 Assessment Of The Paper:**

More detailed information regarding each of these aspects is given below:

**Q2(4) Quality Of Experiments (Optional):**

4: Excellent: The experimental evaluation is comprehensive and the results are compelling.

**Q2(5) Reproducibility:**

3: Good: Key resources (e.g., proofs, code, data) are available and key details (e.g., proofs, experimental setup) are sufficiently well-described for competent researchers to confidently reproduce the main results.

**Q3 Main Strengths:**

Interesting and timely problem. DP-noise adds bias and has to be somehow corrected in the downstream tasks, at least it is intuitively clear that the noise effect in the bias should be removable.

Very well written, strong theory in the logistic regression part especially. I feel that the unbiased alternative proposed in Section 3.3 (PRIVATISING LOGISTIC REGRESSION) is the strongest part of the paper.

**Q4 Main Weakness:**


Somehow it feels that the neural network part falls a bit short. Simply modifying DP-SGD in a way that Poisson subsampling is performed to the joint set of public and private data, and no noise is added corresponding to the public data gradients, is quite obvious solution and I feel there would be more to say about it.

There is something strange with the (eps,delta)-values in the experiments (more details below).

**Q5 Detailed Comments To The Authors:**


The reporting of deltas seems somehow unconventional. Why to use that e^{-6}? For example, in the experiments of Table 3, you say that $\delta = 60000^{-1} - e^{-6}$ but $60000^{-1} - e^{-6} \approx -0.0025$ which does not make sense. Could you comment on this?

**Q7 Justification For Your Score:**

I think is very well written paper with both strong experiments and theory (Section 3.3), and the problem is important (DP synthetic data), so I think it deserves a publication.

**Q9 Complying With Reviewing Instructions:**

1: Yes.

---

### Decision · Program_Chairs · 2022-05-15

**Decision:**

Accept (Oral)

**Comment:**

Meta Review:
This paper proposes a sampling approach to correct bias in the synthetically generated data from a differentially private algorithm. Reviewers are in unanimous agreement that this paper has high technical quality and can be a significant contribution to the community.